# Boundary overlap in the open XXZ spin chain

Charbel ABETIAN[1], Nikolai KITANINE [2*] and Veronique TERRAS [1]

**1** Université Paris-Saclay, CNRS, LPTMS, 91405, Orsay, France
**2** Institut de Mathématiques de Bourgogne, UMR 5584, CNRS and Université de Bourgogne, F-21000 Dijon, France
* Nikolai.Kitanine@u-bourgogne.fr

## Abstract

In this paper we compute the overlaps of the ground states for the open spin chains after a change of one of the boundary magnetic fields. It can be considered as the first step toward the study of the boundary quench problem: behaviour of an open spin chain after an abrupt change of one boundary magnetic field.

# 1   Introduction

Spin chains with open boundaries [1] are interesting example of integrable systems interacting with their environment: the interaction is in that case implemented through the boundary magnetic fields acting on the first and the last site of the chain. It is therefore important to understand what happens if we change one of these boundary magnetic fields. In particular, it can be interesting to study the dynamics after an abrupt change of the magnetic field (boundary quench), or to consider a time-dependent magnetic field (boundary driven system). Although it is a priori only a local perturbation, it was observed that a change of the boundary magnetic field can induce a macroscopic change of the ground state [2] or a phase transition [3].

Integrability gives a unique possibility to study such systems in details. However, a necessary condition to deal with such kind of quench problems in the framework of quantum integrability is the possibility to compute overlaps [4] between the eigenstates before and after the quench. In this paper, we propose the first step in this direction: we compute the overlaps between the ground states of the chain for different values of one the boundary magnetic field.

Typically, being able to exactly calculate the relevant overlaps is crucial to making the study of quench dynamics possible [5–7]. This is often a complicated problem, and the main difficulty comes in general from the fact that the eigenstates before and after the quench are constructed using different algebras: this is for instance the case if the quench comes from a change of the coupling constant. For a boundary quench in which only one boundary magnetic field is modified, however, we can use the same reflexion algebra [8] to describe the eigenstates before and after the quench. This gives a possibility to express overlaps by means of the open chain version of Slavnov formula for the scalar products [9–12].

We consider in this paper the XXZ spin chain in the massive antiferromagnetic regime with diagonal boundary terms (i.e. boundary magnetic fields parallel to the $z$-anisotropy direction). It was shown [2, 13] that low energy properties of the model in this case are determined by two boundary magnetic fields and changing one of them can change the ground state in a macroscopic way. In the diagonal boundary terms case, the eigenstates can be obtained from the boundary version of the Algebraic Bethe Ansatz (ABA) [1], and the Slavnov formula for the overlaps leads to a determinant expression in which the size of the matrix is proportional to the length $L$ of the spin chain [12].

To compute the thermodynamic limit of this determinant representation for the overlaps, we use the method first introduced in [14] which we call *Gaudin extraction*: it permits to express the ratio of Slavnov and Gaudin determinants as a Cauchy determinant (up to an exponentially small correction in $L$). The resulting Cauchy determinant can be easily computed as a product over the ground state Bethe roots.

Using the ground state configuration of the Bethe roots [2, 15], we obtain in the thermodynamic limit a very simple expression for the overlap in terms of the anisotropy parameter and of the two values of the boundary magnetic field. We consider here two possible configurations of Bethe roots: (1) the ground state Bethe roots are all real or (2) there is one complex Bethe root (boundary string). In both configurations there are no real holes, the solutions with holes

(corresponding to excited states) will be treated elsewhere. Our final result is valid in the half-infinite chain limit, however we are able to control the finite size corrections to this result, and to show that all corrections are exponentially small in $L$. We also observe numerically a very rapid convergence towards our analytic result (see appendix A).

We would like to mention that similar results [16, 17] in the massive case can be obtained from the $q$-vertex operator approach [18–20]. It is noteworthy that, in general, the Gaudin extraction method introduced in [14] permits to reproduce in the ABA framework the results for the form factors obtained from the $q$-vertex operator approach. In particular, it permits to relate two possible descriptions for the excited states.

The paper is organised as follows. In section 2, we give a brief technical introduction to the computation of boundary overlaps in the ABA framework: we remind the construction of eigenstates, the structure of the ground state Bethe roots according to the different values of the boundary magnetic fields, and the Slavnov determinant formula for the scalar products. In section 3, we express the overlap as a ratio of determinants and show how to apply to this ratio the Gaudin extraction technique leading to a Cauchy determinant representation. Section 4 is devoted to the computation of the overlaps in the thermodynamic limit: we study separately the case where all Bethe roots are real and the case of a presence of a boundary complex root. Our final result, concerning the thermodynamic limit of the overlap between ground states before and after a boundary quench, is presented in section 5. In the Appendix A we illustrate the rapid convergence of numerical results to our analytic formula for the overlap.

## 2 The open XXZ spin chain with longitudinal boundary fields

We consider here the open XXZ spin-1/2 chain with longitudinal boundary fields. The Hamiltonian of this model is

$$\mathbf{H} \equiv \mathbf{H}_{h^-,h^+} = \sum_{i=1}^{L-1} \left[ \sigma_i^x \sigma_{i+1}^x + \sigma_i^y \sigma_{i+1}^y + \Delta(\sigma_i^z \sigma_{i+1}^z - 1) \right] + h^- \sigma_1^z + h^+ \sigma_N^z, \tag{1}$$

in which $\sigma_i^\alpha$, $\alpha = x, y, z$, denote the Pauli matrices at site $i$, $\Delta$ is the anisotropy of the coupling along the $z$-direction, and $h_-, h_+$ are the values of the boundary fields. We restrict our study to the case in which these boundary fields are oriented along the direction $z$ of the anisotropy, and to the antiferromagnetic regime for which $\Delta > 1$. In the following, we shall parametrize the anisotropy parameter $\Delta$ as

$$\Delta = \frac{q + q^{-1}}{2} = \cosh \zeta, \qquad q = e^{-\zeta}, \qquad \zeta > 0, \tag{2}$$

and the boundary fields $h^\pm$ as

$$h^\pm = -\sinh \zeta \, \coth \xi^\pm. \tag{3}$$

### 2.1 Solution of the model by algebraic Bethe Ansatz

This model is integrable. It can be solved by a boundary version of the algebraic Bethe Ansatz [1], from the construction of a boundary monodromy matrix $\mathcal{U}(\lambda)$. The latter is a $2 \times 2$ matrix of quantum operators depending on a spectral parameter $\lambda$:

$$\mathcal{U}(\lambda) = \begin{pmatrix} \mathcal{A}(\lambda) & \mathcal{B}(\lambda) \\ \mathcal{C}(\lambda) & \mathcal{D}(\lambda) \end{pmatrix}, \qquad \mathcal{A}(\lambda), \, \mathcal{B}(\lambda), \, \mathcal{C}(\lambda), \, \mathcal{D}(\lambda) \in \text{End} \, \mathbb{H}, \tag{4}$$

where $\mathbb{H} = \otimes_{n=1}^{L} \mathbb{H}_n$, $\mathbb{H}_n = \mathbb{C}^2$, is the $2^L$-dimensional quantum space of states of the model. It satisfies, with the 6-vertex R-matrix,

$$
R(\lambda) = \begin{pmatrix} \sin(\lambda - i\zeta) & 0 & 0 & 0 \\ 0 & \sin(\lambda) & \sin(-i\zeta) & 0 \\ 0 & \sin(-i\zeta) & \sin(\lambda) & 0 \\ 0 & 0 & 0 & \sin(\lambda - i\zeta) \end{pmatrix},
\tag{5}
$$

the reflexion equation introduced in [21]:

$$
R(\lambda - \mu)(\mathcal{U}^t(-\lambda) \otimes \mathrm{Id}_2)R(\lambda + \mu + i\zeta)(\mathrm{Id}_2 \otimes \mathcal{U}^t(-\mu))
$$
$$
= (\mathrm{Id}_2 \otimes \mathcal{U}^t(-\mu))R(\lambda + \mu + i\zeta)(\mathcal{U}^t(-\lambda) \otimes \mathrm{Id}_2)R(\lambda - \mu). \quad (6)
$$

Here, $\mathrm{Id}_2$ stands for the $2 \times 2$ identity matrix, and $\mathcal{U}^t$ should be understood as the transposed of $\mathcal{U}$ as a $2 \times 2$ matrix. The reflexion equation (6) gives the commutation relations for the operators $\mathcal{A}(\lambda)$, $\mathcal{B}(\lambda)$, $\mathcal{C}(\lambda)$, $\mathcal{D}(\lambda)$ elements of the matrix (4).

The boundary monodromy matrix $\mathcal{U}(\lambda)$ can be constructed from simpler scalar solutions of (6), the boundary K-matrices. Let us introduce the following diagonal scalar solution of (6),

$$
K(\lambda; \xi) = \begin{pmatrix} \sin(\lambda + i\zeta/2 + i\xi) & 0 \\ 0 & \sin(i\xi - \lambda - i\zeta/2) \end{pmatrix},
\tag{7}
$$

and define

$$
K^-(\lambda) = K(\lambda; \xi^-), \qquad K^+(\lambda) = K(\lambda - i\zeta; \xi^+),
\tag{8}
$$

where $\xi^{\pm}$ parametrize the boundary fields at both ends of the chain, see (3). Let us also introduce the bulk monodromy matrix as the following product of R-matrices along the chain:

$$
T(\lambda) \equiv T_0(\lambda) = R_{0L}(\lambda - \zeta_L) \ldots R_{01}(\lambda - \zeta_1).
\tag{9}
$$

Here, the notation $R_{0n}$, for $1 \le n \le L$, means that the corresponding R-matrix acts on $\mathbb{V}_0 \otimes \mathbb{H}_n$, where $\mathbb{H}_n$ is the quantum space at site $n$, whereas $\mathbb{V}_0 = \mathbb{C}^2$ is a 2-dimensional auxiliary space, so that $T(\lambda)$ can be considered as $2 \times 2$ matrix (on the auxiliary space $\mathbb{V}_0$) of quantum operators. The parameters $\zeta_1, \ldots, \zeta_L$ are inhomogeneity parameters which may be introduced for convenience. Then the boundary monodromy matrix can be constructed from $K^+$ and $T$ as

$$
\mathcal{U}^t(\lambda) \equiv \mathcal{U}_0^{t_0}(\lambda) = (-1)^L T_0^{t_0}(\lambda) \left[ K_0^+ \right]^{t_0}(\lambda) \left[ \sigma_0^y T_0^{t_0}(-\lambda) \sigma_0^y \right]^{t_0}.
\tag{10}
$$

Taking the trace, on the auxiliary space $\mathbb{V}_0$, of the product of this monodromy matrix with the other boundary matrix $K^-$, one obtains a one-parameter family of commuting transfer matrices, which are generating functions of the commuting conserved charges of the model:

$$
\mathcal{T}(\lambda) = \mathrm{tr}_0 \left[ K_0^-(\lambda) \mathcal{U}_0(\lambda) \right].
\tag{11}
$$

The Hamiltonian (1) of the open spin-1/2 chain can then be expressed in terms of this transfer matrix in the homogeneous limit $\zeta_\ell = -i\zeta/2$, $\ell = 1, \ldots, L$, as

$$
\mathbf{H} = \frac{-i \sinh \zeta}{\mathcal{T}(\lambda)} \frac{d}{d\lambda} \mathcal{T}(\lambda) \Big|_{\lambda = -i\zeta/2} + \frac{1}{\cosh \zeta} - 2L \cosh \zeta.
\tag{12}
$$

In this algebraic framework, the common eigenstates of the transfer matrices (and of the Hamiltonian in the homogeneous limit) can be constructed as Bethe states, i.e. by using the

operators $\mathcal{B}(\lambda)$ and $\mathcal{C}(\lambda)$ of (4) as generalised creation and annihilation operators when acting on a reference state. Let us define $|0\rangle$ (respectively $\langle 0|$) to be the reference state (respectively the dual reference state) corresponding to the ferromagnetic state where all spins are up, and let us consider, for a given set of spectral parameters $\{\lambda\} \equiv \{\lambda_1, \ldots, \lambda_N\}$, Bethe states of the form

$$|\{\lambda\}\rangle = \prod_{j=1}^{N} \mathcal{B}(\lambda_j) |0\rangle, \qquad \langle\{\lambda\}| = \langle 0| \prod_{j=1}^{N} \mathcal{C}(\lambda_j). \tag{13}$$

Then, the Bethe states (13) are eigenstates of the transfer matrix $\mathcal{T}(\nu)$ (11), with eigenvalue

$$\tau(\nu, \{\lambda\}) = \mathbf{a}(\nu) \frac{\sin(2\nu - i\zeta)}{\sin(2\nu)} \prod_{i=1}^{N} \frac{\mathfrak{s}(\nu + i\zeta, \lambda_i)}{\mathfrak{s}(\nu, \lambda_i)} + \mathbf{a}(-\nu) \frac{\sin(2\nu + i\zeta)}{\sin(2\nu)} \prod_{i=1}^{N} \frac{\mathfrak{s}(\nu - i\zeta, \lambda_i)}{\mathfrak{s}(\nu, \lambda_i)}. \tag{14}$$

if the corresponding set of parameters $\{\lambda\}$ satisfies the system of Bethe equations

$$\mathbf{a}(\lambda_j) \frac{\sin(2\lambda_j - i\zeta)}{\sin(2\lambda_j)} \prod_{k=1}^{N} \mathfrak{s}(\lambda_j + i\zeta, \lambda_k) + \mathbf{a}(-\lambda_j) \frac{\sin(2\lambda_j + i\zeta)}{\sin(2\lambda_j)} \prod_{k=1}^{N} \mathfrak{s}(\lambda_j - i\zeta, \lambda_k) = 0, \tag{15}$$

for $j = 1, \ldots, N$. Here we have used the shortcut notation

$$\mathfrak{s}(\lambda, \mu) = \sin(\lambda + \mu) \sin(\lambda - \mu) = \sin^2 \lambda - \sin^2 \mu. \tag{16}$$

and defined

$$\mathbf{a}(\lambda) = (-1)^L a(\lambda) d(-\lambda) \sin(\lambda + i\xi^+ + i\zeta/2) \sin(\lambda + i\xi^- + i\zeta/2), \tag{17}$$

with

$$a(\lambda) = \prod_{\ell=1}^{L} \sin(\lambda - \zeta_\ell - i\zeta), \qquad d(\lambda) = \prod_{\ell=1}^{L} \sin(\lambda - \zeta_\ell). \tag{18}$$

Bethe states of the form (13) are called on-shell Bethe states if the corresponding set of parameters $\{\lambda\}$ satisfies the Bethe equations (15), and off-shell Bethe states otherwise. On-shell Bethe states are therefore eigenstates of the transfer matrix (11). They are also eigenstates of the Hamiltonian (1) in the homogeneous limit $\zeta_\ell = -i\zeta/2$, $\ell = 1, \ldots, L$. The corresponding energy is

$$E(\{\lambda\}) = h^+ + h^- - \sum_{j=1}^{N} \frac{4\sinh^2 \zeta}{\cosh \zeta - \cos(2\lambda_j)}. \tag{19}$$

In the following, it will be convenient to rewrite the Bethe equations (15) as[1]

$$\mathfrak{a}(\lambda_j|\{\lambda\}) = 1, \qquad 1 \le j \le N, \tag{20}$$

by introducing the exponential counting function

$$\mathfrak{a}(\nu|\{\lambda\}) = \frac{\mathbf{a}(\nu)}{\mathbf{a}(-\nu)} \frac{\sin(i\zeta - 2\nu)}{\sin(i\zeta + 2\nu)} \prod_{\ell=1}^{N} \frac{\mathfrak{s}(\nu + i\zeta, \lambda_\ell)}{\mathfrak{s}(\nu - i\zeta, \lambda_\ell)}. \tag{21}$$

In terms of this exponential counting function, the eigenvalue (14) of the transfer matrix can be expressed as

$$\tau(\nu, \{\lambda\}) = -\mathbf{a}(-\nu) \frac{\sin(2\nu + i\zeta)}{\sin(2\nu)} \prod_{i=1}^{N} \frac{\mathfrak{s}(\nu - i\zeta, \lambda_i)}{\mathfrak{s}(\nu, \lambda_i)} \left[ \mathfrak{a}(\nu|\{\lambda\}) - 1 \right]. \tag{22}$$

---

[1]In this rewriting, one should exclude the root $0, \frac{\pi}{2}$ of the function $\mathfrak{a}(u|\{\lambda\}) - 1$.

## 2.2 Scalar products of Bethe states

In this algebraic framework, one can express scalar products between an on-shell and an off-shell Bethe vectors of the form (13) in terms of a determinant of usual functions, see [11, 12]. This representation is the analog, in the open case, of the determinant representation obtained in [9] in the periodic case. We recall it here.

For $\{\lambda\} \equiv \{\lambda_1, \ldots, \lambda_N\}$ a solution of the Bethe equations and $\{\mu\} \equiv \{\mu_1, \ldots, \mu_N\}$ an arbitrary set of parameters, the scalar product $\langle\{\lambda\}|\{\mu\}\rangle$ is equal to the scalar product $\langle\{\mu\}|\{\lambda\}\rangle$, and admits the following determinant representation:

$$
\begin{aligned}
\langle\{\lambda\}|\{\mu\}\rangle &= \langle\{\mu\}|\{\lambda\}\rangle \\
&= \prod_{j=1}^{N}\left[(-1)^L a(\lambda_j) d(-\lambda_j) \frac{\sin(2\lambda_j - i\zeta)\sin(2\mu_j - i\zeta)}{\sin(2\mu_j)} \frac{\sin(\lambda_j + i\xi_+ + i\frac{\zeta}{2})}{\sin(\lambda_j - i\xi_- - i\frac{\zeta}{2})}\right] \\
&\quad \times \prod_{j<k}\left[\frac{\sin(\lambda_j + \lambda_k - i\zeta)}{\sin(\lambda_j + \lambda_k + i\zeta)} \frac{1}{\mathfrak{s}(\lambda_j,\lambda_k)\mathfrak{s}(\mu_k,\mu_j)}\right] \det_N\left[H(\boldsymbol{\lambda},\boldsymbol{\mu})\right],
\end{aligned} \tag{23}
$$

where the elements of the $N \times N$ matrix $H(\boldsymbol{\lambda},\boldsymbol{\mu})$ are

$$
\left[H(\boldsymbol{\lambda},\boldsymbol{\mu})\right]_{jk} = \frac{\sin(-i\zeta)}{\mathfrak{s}(\mu_k,\lambda_j)}\left[\mathbf{a}(\mu_k)\prod_{\ell\neq j}\mathfrak{s}(\mu_k + i\zeta, \lambda_\ell) - \mathbf{a}(-\mu_k)\prod_{\ell\neq j}\mathfrak{s}(\mu_k - i\zeta, \lambda_\ell)\right], \tag{24}
$$

for $\boldsymbol{\lambda} \equiv (\lambda_1, \ldots, \lambda_N)$ and $\boldsymbol{\mu} \equiv (\mu_1, \ldots \mu_N)$. When $\{\lambda\}$ and $\{\mu\}$ coincide, (23) becomes

$$
\begin{aligned}
\langle\{\lambda\}|\{\lambda\}\rangle &= \prod_{j=1}^{N}\left[(-1)^L a(\lambda_j) d(-\lambda_j) \sin(2\lambda_j - i\zeta) \frac{\sin(\lambda_j + i\xi_+ + i\frac{\zeta}{2})}{\sin(\lambda_j - i\xi_- - i\frac{\zeta}{2})}\right] \\
&\quad \times \prod_{j<k}\frac{\sin(\lambda_j + \lambda_k - i\zeta)}{\sin(\lambda_j + \lambda_k + i\zeta)}\prod_{k=1}^{N}\frac{\mathbf{a}(-\lambda_k)\prod_{\ell=1}^{N}\mathfrak{s}(\lambda_k - i\zeta, \lambda_\ell)}{i\sin^2(2\lambda_k)\prod_{\ell\neq k}\mathfrak{s}(\lambda_k,\lambda_\ell)}\det_N\left[\mathcal{M}(\boldsymbol{\lambda},\boldsymbol{\lambda})\right], \tag{25}
\end{aligned}
$$

where the elements of the $N \times N$ matrix $\mathcal{M}(\boldsymbol{\lambda},\boldsymbol{\lambda})$ are

$$
\left[\mathcal{M}(\boldsymbol{\lambda},\boldsymbol{\lambda})\right]_{jk} = i\,\delta_{jk}\,\mathfrak{a}'(\lambda_j|\{\lambda\}) - 2\pi\left[K(\lambda_j - \lambda_k) - K(\lambda_j + \lambda_k)\right]. \tag{26}
$$

Here we have defined

$$
K(\lambda) = \frac{\sinh(2\zeta)}{2\pi\sin(\lambda + i\zeta)\sin(\lambda - i\zeta)} = \frac{1}{2\pi}\left[t(\lambda) + t(-\lambda)\right], \tag{27}
$$

with

$$
t(v) = \frac{\sinh\zeta}{\sin v \sin(v - i\zeta)}, \tag{28}
$$

and $\mathfrak{a}'(v|\{\lambda\})$ denotes the derivative, with respect to the variable $v$, of the exponential counting function (21).

## 2.3 Description of the ground state

The study of the solutions of Bethe equations in the homogeneous limit $\zeta_\ell = -i\frac{\zeta}{2}$, $\ell = 1, \ldots, L$, and in particular of the ground state of the Hamiltonian (1) in the thermodynamic limit $L \to \infty$, has been performed in [2, 15, 22]. We briefly recall here the results which will be useful for our study. As mentioned above, we restrict ourselves to the antiferromagnetic regime

$\Delta > 1$, with $\Delta$ parametrized as in (2). The boundary fields $h^\sigma$, $\sigma = \pm$, are parameterized as in (3), with

$$\xi^\sigma = -\tilde{\xi}^\sigma + i\delta^\sigma \frac{\pi}{2}, \qquad \tilde{\xi}^\sigma \in \mathbb{R}, \qquad \delta^\sigma = \begin{cases} 1 & \text{if } |h^\sigma| < \sinh\zeta, \\ 0 & \text{if } |h^\sigma| > \sinh\zeta. \end{cases} \tag{29}$$

Due to the parity and periodicity of the Bethe equations, it is enough to consider solutions of (20) such that $0 < \Re(\lambda_j) < \frac{\pi}{2}$ or $(\Re(\lambda_j) = 0, \frac{\pi}{2}$ and $\Im(\lambda_j) < 0)$. Low-energy states are given by solutions of (20) close to half-filling for which nearly all Bethe roots are real. Complex roots appear by complex conjugated pairs $\lambda_j, \bar{\lambda}_j$, except if $\Re(\lambda_j) = 0, \frac{\pi}{2}$.

The Bethe equations for real roots can be rewritten in logarithmic form as

$$\hat{\xi}(\lambda_j | \{\lambda\}) = \frac{\pi n_j}{L}, \qquad n_j \in \mathbb{Z}, \tag{30}$$

in which $\hat{\xi}(\mu | \{\lambda\}) = -\frac{i}{2L} \log \mathfrak{a}(\mu | \{\lambda\})$ is the counting function:

$$\hat{\xi}(\mu | \{\lambda\}) = p(\mu) + \frac{g(\mu)}{2L} - \frac{\theta(2\mu)}{2L} + \frac{1}{2L} \sum_{k=1}^{N} \left[\theta(\mu - \lambda_k) + \theta(\mu + \lambda_k)\right]. \tag{31}$$

The functions $p, \theta$ and $g$ appearing in (31) are defined on the real axis by

$$p(\lambda) = i \log \frac{\sin(i\zeta/2 + \lambda)}{\sin(i\zeta/2 - \lambda)}, \qquad \theta(\lambda) = i \log \frac{\sin(i\zeta - \lambda)}{\sin(i\zeta + \lambda)}, \tag{32}$$

$$g(\lambda) = i \log \prod_{\sigma = \pm} \frac{\sin(\lambda - i\xi^\sigma - i\zeta/2)}{\sin(\lambda + i\xi^\sigma + i\zeta/2)}, \tag{33}$$

and by appropriate analytical continuation around this axis.

In the thermodynamic limit, the real Bethe roots for the ground state (and more generally for low-energy states) form a dense distribution on the interval $(0, \frac{\pi}{2})$, which can be extended by parity on the interval $(-\frac{\pi}{2}, \frac{\pi}{2})$. The corresponding Bethe equations (30) turn at the leading order into an integral equation for their density $\rho(\lambda)$,

$$\rho(\lambda) + \int_{-\frac{\pi}{2}}^{\frac{\pi}{2}} K(\lambda - \beta) \rho(\beta) \, d\beta = \frac{p'(\lambda)}{\pi}, \tag{34}$$

with solution

$$\rho(\lambda) = \frac{1}{\pi} \frac{\vartheta_1'(0)}{\vartheta_2(0)} \frac{\vartheta_3(\lambda)}{\vartheta_4(\lambda)}. \tag{35}$$

Here and in the following, $\vartheta_i(\lambda) \equiv \vartheta_i(\lambda, q)$, $i \in \{1, 2, 3, 4\}$, denote the Theta functions of nome $q$ defined as in [23]. The function $p'$ in (34) is the derivative of the function $p$ (32) and corresponds, up to a shift in the argument, to the function $t$ (28), whereas the function $K$ (27) corresponds, up to a scalar factor, to the derivative of the function $\theta$:

$$p'(\lambda) = t(\lambda + i\tfrac{\zeta}{2}), \qquad K(\lambda) = -\frac{1}{2\pi} \theta'(\lambda). \tag{36}$$

A detailed analytical description of the ground state was performed in [2], and different regimes can be distinguished according to the values of $h^+$ and $h^-$ with respect to the two boundary critical fields $h_{\text{cr}}^{(1)}$ and $h_{\text{cr}}^{(2)}$ defined as [15,19]

$$h_{\text{cr}}^{(1)} = \Delta - 1, \qquad h_{\text{cr}}^{(2)} = \Delta + 1. \tag{37}$$

In particular, for

$$h^-, h^+ > h_{\mathrm{cr}}^{(1)} \quad \text{or} \quad -h^-, -h^+ > h_{\mathrm{cr}}^{(1)}, \qquad \text{if } L \text{ is even,} \tag{38}$$

or for

$$h^-, -h^+ > h_{\mathrm{cr}}^{(1)} \quad \text{or} \quad -h^-, h^+ > h_{\mathrm{cr}}^{(1)}, \qquad \text{if } L \text{ is odd,} \tag{39}$$

the spectrum becomes gapless in the thermodynamic limit. In the following, we shall restrict our study to values of the boundary fields for which the spectrum remains gapped in the thermodynamic limit, which means that we exclude the consideration of the configurations (38) and (39). We therefore have to distinguish the following different cases [2]:

- **For $L$ even:**

  (A)  $|h^{\sigma_1}| < h_{\mathrm{cr}}^{(1)}$ with $h^{\sigma_1} > h^{\sigma_2}$ ($\{\sigma_1, \sigma_2\} = \{+, -\}$).

  The ground state is in the sector $N = \frac{L}{2}$ (with magnetization 0). It is given by $\frac{L}{2} - 1$ real roots with adjacent quantum numbers $n_j = 1, \ldots, \frac{L}{2} - 1$ and an isolated complex root of the form

  $$\lambda_{\mathrm{BR}}^{\sigma_1} = -i(\zeta/2 + \xi^{\sigma_1} + \epsilon^{\sigma_1}) = -i(\zeta/2 - \tilde{\xi}^{\sigma_1} + \epsilon^{\sigma_1}) + \delta^{\sigma_1} \frac{\pi}{2}, \tag{40}$$

  where $\epsilon^{\sigma_1}$ is a finite length correction which becomes exponentially small for large $L$, i.e. $\epsilon^{\sigma_1} = O(L^{-\infty})$.

  (B)  $h^{\sigma_2} < h_{\mathrm{cr}}^{(1)} < h^{\sigma_1} < h_{\mathrm{cr}}^{(2)}$ ($\{\sigma_1, \sigma_2\} = \{+, -\}$).

  The ground state is in the sector $N = \frac{L}{2}$ (with magnetization 0). It is given by $\frac{L}{2}$ real roots with adjacent quantum numbers $n_j = 1, \ldots, \frac{L}{2}$.

  (C)  $h^{\sigma_2} < h_{\mathrm{cr}}^{(1)} < h_{\mathrm{cr}}^{(2)} < h^{\sigma_1}$ ($\{\sigma_1, \sigma_2\} = \{+, -\}$).

  The ground state is in the sector $N = \frac{L}{2}$ (with magnetization 0). It is given by $\frac{L}{2} - 1$ real roots with adjacent quantum numbers $n_j = 1, \ldots, \frac{L}{2} - 1$ and an isolated complex root of the form (40).

- **For $L$ odd:**

  (A')  $h^{\pm} < h_{\mathrm{cr}}^{(1)}$ with $h^+ + h^- < 0$.

  The ground state is in the sector $N = \frac{L-1}{2}$ (with magnetization $+1/2$). It is given by $N = \frac{L-1}{2}$ real Bethe roots with adjacent quantum numbers.

  (B')  $h^{\pm} > -h_{\mathrm{cr}}^{(1)}$ with $h^+ + h^- > 0$.

  The ground state is in the sector $N = \frac{L+1}{2}$ (with magnetization $-1/2$). Computations of physical quantities in this case can simply be obtained from the previous case (A') by using the invariance of the model under the reversal of all spins together with a change of sign of the boundary fields $h^{\pm}$.

Note that there are no holes in the above configurations of Bethe roots (A) to (B').

Computations of physical quantities in the case (C) can also be obtained from cases (A) and (B) by using the invariance of the model under the reversal of all spins together with a change of sign of the boundary fields $h^{\pm}$.

As in [2], the isolated complex root of the form (40) will be called *boundary root* (it was called *boundary 1-string* in [15]). It was shown in [2] that such a boundary root plays an important role in the computation of physical quantities, such as the boundary magnetization at the thermodynamic limit.

## 2.4 Changing the boundary field $h_-$

In the following, we shall consider the effects of a change of the boundary field $h_-$ in the Hamiltonian (1). Hence, we consider a local quench in which we pass from a Hamitonian $\mathbf{H}_1$ with left boundary field $h_1^-$ to a Hamiltonian $\mathbf{H}_2$ with left boundary field $h_2^-$, the other parameters $\Delta$ and $h^+$ of the Hamiltonian remaining unchanged.

It is important to remark that such a quench leaves the boundary monodromy matrix (4) unchanged, since the latter do not depend on $h^-$, see (10). Hence, the Bethe states take the same algebraic form (13) for both Hamiltonians. It is therefore possible to compute their overlap by using the determinant representation (23). This is the purpose of the next sections. In particular, our aim is to compute the normalized overlap between the ground state $|\{\lambda\}\rangle$, with Bethe roots that we shall denote by $\{\lambda\}$, of the Hamiltonian $\mathbf{H}_1$, and the ground state $|\{\mu\}\rangle$, with Bethe roots that we shall denote by $\{\mu\}$, of the Hamiltonian $\mathbf{H}_2$.

In the following, we shall adopt a subscript 1, respectively a subscript 2, for all quantities which depend on the boundary field $h_1^-$ (parametrized by $\xi_1^-$) of the Hamiltonian $\mathbf{H}_1$, respectively on the boundary field $h_2^-$ (parametrized by $\xi_2^-$) of the Hamiltonian $\mathbf{H}_2$. In particular, the Bethe equations for the ground state of $\mathbf{H}_1$ are

$$\mathfrak{a}_1(\lambda_j|\{\lambda\}) = 1, \qquad j = 1, \ldots, N_1, \tag{41}$$

whereas the Bethe equations for the ground state of $\mathbf{H}_2$ are

$$\mathfrak{a}_2(\mu_j|\{\mu\}) = 1, \qquad j = 1, \ldots, N_2. \tag{42}$$

The respective transfer matrix eigenvalues will be denoted by $\tau_1(\nu|\{\lambda\})$ and $\tau_2(\nu|\{\mu\})$.

## 3 Cauchy determinant representation for the overlap

The main goal of this section is to provide an exact representation for the overlap between the ground states of the two Hamiltonians $\mathbf{H}_1$ and $\mathbf{H}_2$, in the form of a generalised Cauchy determinant. More precisely, we consider the following normalized version of the overlap:

$$S(\{\lambda\}, \{\mu\}) = \frac{\langle\{\lambda\}|\{\mu\}\rangle}{\langle\{\lambda\}|\{\lambda\}\rangle} \frac{\langle\{\mu\}|\{\lambda\}\rangle}{\langle\{\mu\}|\{\mu\}\rangle}. \tag{43}$$

We recall that $\{\lambda\} = \{\lambda_1, \ldots, \lambda_{N_1}\}$, respectively $\{\mu\} = \{\mu_1, \ldots, \mu_{N_2}\}$, denotes the set of Bethe roots corresponding to the ground state of the Hamiltonian $\mathbf{H}_1$ with first site boundary field $h_1^-$, respectively of the Hamiltonian $\mathbf{H}_2$ with first site boundary field $h_2^-$. Note that the overlap (43) vanishes identically if $N_1 \neq N_2$, so that we restrict ourselves to cases for which the two ground states have the same number of Bethe roots $N_1 = N_2 = N$.

### 3.1 Determinant representation

We recall that, since the Hamiltonians $\mathbf{H}_1$ and $\mathbf{H}_2$ differ only by the value of the boundary field $h^-$ at first site (the boundary field $h^+$ remaining the same for both Hamiltonians), they share the same boundary monodromy matrix (4). Hence, the scalar products appearing in (43) can be represented by means of the formulas of section 2.2. More precisely, representing the first ratio of (43) by means of (23) and (25), and the second ratio by means of similar formulas in which we interchange the role of $\{\lambda\}$ and $\{\mu\}$, we obtain, after some slight simplifications:

$$S(\{\lambda\},\{\mu\}) = \prod_{k=1}^{N} \frac{\mathbf{a}_1(-\mu_k)}{\mathbf{a}_2(-\mu_k)} \frac{\mathbf{a}_2(-\lambda_k)}{\mathbf{a}_1(-\lambda_k)} \prod_{k=1}^{N} \prod_{l=1}^{N} \frac{\mathfrak{s}(\mu_k - i\zeta, \lambda_l)\,\mathfrak{s}(\lambda_k - i\zeta, \mu_l)}{\mathfrak{s}(\lambda_k - i\zeta, \lambda_l)\,\mathfrak{s}(\mu_k - i\zeta, \mu_l)}$$

$$\times \frac{\det_N \hat{H}_1(\boldsymbol{\lambda},\boldsymbol{\mu})}{\det_N \mathcal{M}_1(\boldsymbol{\lambda},\boldsymbol{\lambda})} \frac{\det_N \hat{H}_2(\boldsymbol{\mu},\boldsymbol{\lambda})}{\det_N \mathcal{M}_2(\boldsymbol{\mu},\boldsymbol{\mu})}. \quad (44)$$

Here we have defined, for two $N$-tuples $\boldsymbol{\nu} = (\nu_1,\ldots,\nu_N)$ and $\boldsymbol{\omega} = (\omega_1,\ldots,\omega_N)$ of complex numbers, the matrix $\hat{H}(\boldsymbol{\nu},\boldsymbol{\omega})$ with elements

$$\left[\hat{H}(\boldsymbol{\nu},\boldsymbol{\omega})\right]_{jk} = \mathfrak{a}(\omega_k|\{\nu\})\left[t(-\omega_k + \nu_j) - t(-\omega_k - \nu_j)\right] + t(\omega_k - \nu_j) - t(\omega_k + \nu_j), \quad (45)$$

in terms of the function (28). As mentioned in section (2.4), the index 1, respectively 2, means that the corresponding quantity depends explicitly on the boundary field $h_1^-$, respectively $h_2^-$. In other words,

$$\left[\hat{H}_\ell(\boldsymbol{\nu},\boldsymbol{\omega})\right]_{jk} = \mathfrak{a}_\ell(\omega_k|\{\nu\})\left[t(-\omega_k + \nu_j) - t(-\omega_k - \nu_j)\right] + t(\omega_k - \nu_j) - t(\omega_k + \nu_j), \quad (46)$$

for $\ell = 1,2$, with $\mathfrak{a}_\ell$ being the exponential counting function (21) expressed in terms of the function $\mathbf{a}_\ell$ (17) with boundary field $h_\ell^-$ parametrized by $\xi_\ell^-$. Similar notations are used for the Gaudin matrix (26), i.e.

$$[\mathcal{M}_\ell(\boldsymbol{\nu},\boldsymbol{\nu})]_{jk} = i\,\delta_{jk}\,\mathfrak{a}_\ell'(\nu_j|\{\nu\}) - 2\pi\left[K(\nu_j - \nu_k) - K(\nu_j + \nu_k)\right], \quad \ell = 1,2. \quad (47)$$

Note that, to obtain the expression (44) from (23)-(24), we have used the following simple identity:

$$\frac{\sinh\zeta}{\mathfrak{s}(\lambda,\mu)\,\mathfrak{s}(\lambda - i\zeta,\mu)} = \frac{t(\lambda - \mu) - t(\lambda + \mu)}{\sin(2\lambda - i\zeta)\sin(2\mu)}. \quad (48)$$

## 3.2 Gaudin extraction

In this subsection, we explain how to transform the ratios of determinants appearing in (44), adapting to the open case a procedure introduce in [14]. This will enable us to rewrite the overlap in terms of some generalised Cauchy determinant.

Let us consider the following ratio of determinants:

$$\frac{\det_N\left[\hat{H}(\boldsymbol{\nu},\boldsymbol{\omega})\right]}{\det_N[\mathcal{M}(\boldsymbol{\nu},\boldsymbol{\nu})]} = \det_N\left[\mathcal{M}(\boldsymbol{\nu},\boldsymbol{\nu})^{-1}\hat{H}(\boldsymbol{\nu},\boldsymbol{\omega})\right] \quad (49)$$

involving the matrices (45) and (26). Here $\boldsymbol{\nu}$ stands for a $N$-tuple of on-shell Bethe roots, whereas $\boldsymbol{\omega}$ stands, for the moment, for any arbitrary $N$-tuple of complex numbers. Our aim is to compute the elements of the matrix $\mathcal{F}(\boldsymbol{\nu},\boldsymbol{\omega}) = \mathcal{M}(\boldsymbol{\nu},\boldsymbol{\nu})^{-1}\hat{H}(\boldsymbol{\nu},\boldsymbol{\omega})$. The latter are given as the unique solution to the following system of linear equations:

$$i\,\mathfrak{a}'(\nu_j|\{\nu\})[\mathcal{F}(\boldsymbol{\nu},\boldsymbol{\omega})]_{jk} - 2\pi\sum_l\left[K(\nu_j - \nu_l) - K(\nu_j + \nu_l)\right][\mathcal{F}(\boldsymbol{\nu},\boldsymbol{\omega})]_{lk}$$

$$= \mathfrak{a}(\omega_k|\{\nu\})\left[t(-\omega_k + \nu_j) - t(-\omega_k - \nu_j)\right] + t(\omega_k - \nu_j) - t(\omega_k + \nu_j). \quad (50)$$

Following the strategy used in [14], let us look for a function $G(u,w) \equiv G_{\{\nu\}}(u,w)$ of two complex variables $u,w \in D_\zeta = \{z \in \mathbb{C}\,|\,|\Im(z)| < \zeta\}$, solution of the equation

$$G(u,w) + 2\pi i \sum_{\ell} \left[ K(u - v_\ell) - K(u + v_\ell) \right] \frac{G(v_\ell, w)}{\mathfrak{a}'(v_\ell | \{v\})}$$

$$= \mathfrak{a}(w | \{v\}) \left[ t(u - w) - t(-u - w) \right] + t(-u + w) - t(u + w). \quad (51)$$

Note that, if we manage to construct such a solution to (51), it will give the elements of $\mathcal{F}(\boldsymbol{v}, \boldsymbol{\omega})$ by setting

$$i\mathfrak{a}'(v_j | \{v\}) [\mathcal{F}(\boldsymbol{v}, \boldsymbol{\omega})]_{jk} = G(v_j, \omega_k), \quad (52)$$

We shall moreover look for a solution of (51) such that

(i) $u \mapsto G(u, w)$ is a meromorphic function in the strip $D_\zeta$, with no poles at the points $v_\ell, 1 \le \ell \le N$,

(ii) $u \mapsto G(u, w)$ is odd: $G(-u, w) = -G(u, w)$.

Under the assumptions (i) and (ii), (51) can be rewritten as

$$G(u,w) + \oint_{\Gamma_{\boldsymbol{v}, \mathbb{R}}^\pm} K(u - z) \frac{G(z, w)}{\mathfrak{a}(z | \{v\}) - 1} \, dz + 2\pi i \sum_{\ell \in \mathcal{Z}_{\boldsymbol{v}}} \left[ K(u - v_\ell) - K(u + v_\ell) \right] \frac{G(v_\ell, w)}{\mathfrak{a}'(v_\ell | \{v\})}$$

$$= \mathfrak{a}(w | \{v\}) \left[ t(u - w) - t(-u - w) \right] + t(-u + w) - t(u + w), \quad (53)$$

in which the contour $\Gamma_{\boldsymbol{v}, \mathbb{R}}^\pm$ surrounds, with index 1, the elements of the set $(\{v\} \cup \{-v\}) \cap \mathbb{R}$ (and no other poles of the integrand), whereas the remaining sum runs over the set $\mathcal{Z}_{\boldsymbol{v}}$ of indices corresponding to the subset of complex Bethe roots: $\{v_\ell\}_{\ell \in \mathcal{Z}_{\boldsymbol{v}}} = \{v\} \cap (\mathbb{C} \setminus \mathbb{R})$.

It is clear that any function $G(u, w)$ solution of (53) should be a $\pi$-periodic function in $u$: $G(u + \pi, w) = G(u, w)$. Moreover, taking into account the requirement that $u \mapsto G(u, w)$ is a meromorphic function, (53) implies that the only poles of this function that may be on the real axis are those at $u = \pm w \mod \pi$ (if $w \in \mathbb{R}$). The corresponding residues can be easily derived from the R.H.S. of (53):

$$\mathrm{Res} \left[ G(u, w) \right]_{u = \pm w} = i \left( \mathfrak{a}(w | \{v\}) - 1 \right). \quad (54)$$

Hence, for $\epsilon > 0$ and small enough, we can deform the contour $\Gamma_{\boldsymbol{v}, \mathbb{R}}^\pm$ into a rectangle $\Gamma_\epsilon$ with vertices at $(-\pi/2, i\epsilon), (-\pi/2, -i\epsilon), (\pi/2, i\epsilon), (\pi/2, -i\epsilon)$. This can be done provided we sub-

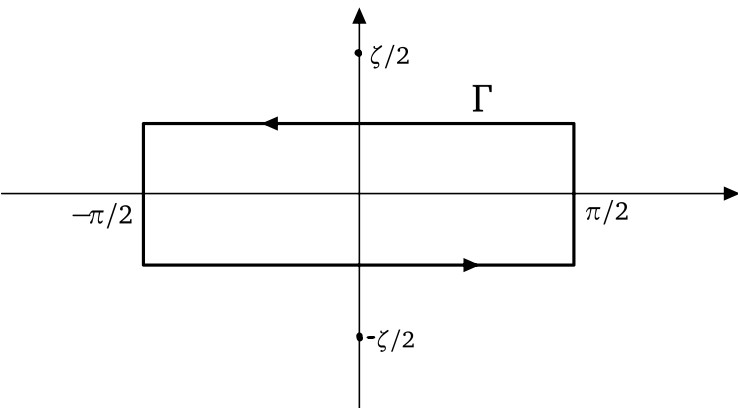

Figure 1: contour $\Gamma_\epsilon$

tract the contribution of the potential other poles that become encircled by the contour. If

there are no holes, i.e. if the function $\mathfrak{a}(u|\{v\}) - 1$ has no other roots on the interval $(0, \frac{\pi}{2})$ than the real Bethe roots $v_j$, the contour integral can be rewritten as

$$\oint_{\Gamma_{v,\mathbb{R}}^{\pm}} K(u-z) \frac{G(z,w)}{\mathfrak{a}(z|\{v\}) - 1} dz = \oint_{\Gamma_\epsilon} K(u-z) \frac{G(z,w)}{\mathfrak{a}(z|\{v\}) - 1} dz$$
$$- 2\pi i \delta_{w\in\mathbb{R}} \left[ iK(u-w) - iK(u+w) \mathfrak{a}(w|\{v\}) \right], \quad (55)$$

in which the notation $\delta_{w\in\mathbb{R}}$ means that the last term is non-zero only if $w \in \mathbb{R}$. Hence (53) can be rewritten as

$$G(u,w) + \oint_{\Gamma_\epsilon} K(u-z) \frac{G(z,w)}{\mathfrak{a}(z|\{v\}) - 1} dz + 2\pi i \sum_{\ell \in \mathcal{Z}_v} \left[ K(u-v_\ell) - K(u+v_\ell) \right] \frac{G(v_\ell, w)}{\mathfrak{a}'(v_\ell|\{v\})}$$
$$= \begin{cases} \left[ t(u-w) + t(u+w) \right]\left[ \mathfrak{a}(w|\{v\}) - 1 \right] & \text{if } w \in \mathbb{R}, \\ \mathfrak{a}(w|\{v\}) \left[ t(u-w) - t(-u-w) \right] + t(-u+w) - t(u+w) & \text{if } w \notin \mathbb{R}. \end{cases} \quad (56)$$

Let us first suppose that all the Bethe roots $v_1, \ldots, v_N$ are real. It is not difficult to see, following standards arguments [2, 24], that the quantity $\mathfrak{a}(w|\{v\})$ has the following behaviour in $L$ for $w$ above and below the real axis:

$$\mathfrak{a}(w|\{v\}) = O(L^{+\infty}) \qquad \text{for} \quad -\zeta < \Im(w) < 0, \qquad (57)$$
$$\mathfrak{a}(w|\{v\}) = O(L^{-\infty}) \qquad \text{for} \quad 0 < \Im(w) < \zeta. \qquad (58)$$

Here and in the following, the notation $O(L^{+\infty})$ (respectively $O(L^{-\infty})$) means that the corresponding quantity is diverging exponentially with $L$ (respectively is vanishing exponentially with $L$). Hence, the equation (56) simplifies drastically for large $L$:

$$G(u,w) + \int_{-\frac{\pi}{2}+i0}^{\frac{\pi}{2}+i0} K(u-z) G(z,w) dz$$
$$= \begin{cases} \left[ t(u-w) + t(u+w) \right]\left[ \mathfrak{a}(w|\{v\}) - 1 \right] + O(L^{-\infty}) & \text{if } w \in \mathbb{R}, \\ t(-u+w) - t(u+w) + O(L^{-\infty}) & \text{if } 0 < \Im(w) < \zeta, \\ \mathfrak{a}(w|\{v\}) \left[ t(u-w) - t(-u-w) + O(L^{-\infty}) \right] & \text{if } -\zeta < \Im(w) < 0. \end{cases} \quad (59)$$

This integral equation has to be compared with the Lieb equation (34). Note that all the functions involved in (34) are holomorphic functions in the strip $D_{\zeta/2} = \{z \in \mathbb{C} \, | \, |\Im(z)| < \zeta/2\}$, so that (34) remains valid in the whole strip $\lambda \in D_{\zeta/2}$. Moreover, the integration contour in (34) can be shifted by a fixed imaginary value as soon as we do not cross poles. Hence, up to exponentially small corrections in $L$, the integral equation (59) admits a solution which is given by a linear combination of solutions of (34):

$$G(u,w) = i\left[ \mathfrak{a}(w|\{v\}) - 1 \right]\left[ \bar{\rho}(u,w) + O(L^{-\infty}) \right], \qquad (60)$$

in which $\bar{\rho}$ is given in term of the function $\rho$ (35) as

$$\bar{\rho}(u,w) = -i\pi \left[ \rho(u-w-i\zeta/2) + \rho(u+w-i\zeta/2) \right]$$
$$= \frac{\vartheta_1'(0)}{\vartheta_2(0)} \left[ \frac{\vartheta_2(u-w)}{\vartheta_1(u-w)} + \frac{\vartheta_2(u+w)}{\vartheta_1(u+w)} \right]. \qquad (61)$$

Note that (60)-(61) provides a solution for all cases considered in (59) thanks to the parity and quasi-periodicity properties of the function $\rho$ (35). Note also that the function (61) satisfies

the properties (i) and (ii), so that it also provides a solution of (51). Hence the elements of the matrix $\mathcal{F}(\boldsymbol{v}, \boldsymbol{\omega}) = \mathcal{M}(\boldsymbol{v}, \boldsymbol{v})^{-1} \hat{H}(\boldsymbol{v}, \boldsymbol{\omega})$ are computed by (52) up to exponentially small corrections in $L$, and

$$\frac{\det_N \left[ \hat{H}(\boldsymbol{v}, \boldsymbol{\omega}) \right]}{\det_N [\mathcal{M}(\boldsymbol{v}, \boldsymbol{v})]} = \det_N [\mathcal{F}(\boldsymbol{v}, \boldsymbol{\omega})] = \prod_{j=1}^{N} \frac{\mathfrak{a}(\omega_j | \{v\}) - 1}{\mathfrak{a}'(v_j | \{v\})} \det_N \left[ \bar{\rho}(v_j, \omega_k) \right] (1 + O(L^{-\infty})). \quad (62)$$

Let us now suppose that the set of Bethe roots $\{v\}$ contains only real roots except for one single boundary complex root $v_{\text{BR}}^{\sigma} = -i(\frac{\zeta}{2} + \xi^{\sigma} + \epsilon^{\sigma})$, $\sigma = \pm$, with $\epsilon^{\sigma} = O(L^{-\infty})$. Then, the large $L$ behaviour (57)-(58) is still valid provided that $w$ remains at finite distance from the zeroes and the poles of the function $\mathfrak{a}$, which we suppose from now on: this is notably the case when $w$ is in the vicinity of the real axis, or when it coincides with a boundary root built on a different boundary parameter as those appearing in $\mathfrak{a}$. The difference with the previous case is that we have a priori to add to the left hand side of (59) an additional term of the form

$$\left[ K(u - v_{\text{BR}}^{\sigma}) - K(u + v_{\text{BR}}^{\sigma}) \right] \frac{G(v_{\text{BR}}^{\sigma}, w)}{\mathfrak{a}'(v_{\text{BR}}^{\sigma} | \{v\})}. \quad (63)$$

Let us however remark that

$$\mathfrak{a}'(v_{\text{BR}}^{\sigma} | \{v\}) = 2Li \, \hat{\xi}'(v_{\text{BR}}^{\sigma} | \{v\}) \underset{L \to \infty}{\sim} i \, g'(v_{\text{BR}}^{\sigma}) = O(\tfrac{1}{\epsilon^{\sigma}}), \quad (64)$$

which diverges exponentially with $L$. This means that the correction induced by the presence of the additional term (63) to the solution of (59) is in fact exponentially small in $L$ with respect to the leading order, i.e. that the solution (60)-(61)-(62) still holds even in the presence of a boundary root.

This result can be applied directly to the computation of the two ratios in (44):

$$\frac{\det_N \hat{H}_1(\boldsymbol{\lambda}, \boldsymbol{\mu})}{\det_N \mathcal{M}_1(\boldsymbol{\lambda}, \boldsymbol{\lambda})} = \prod_{i=1}^{N} \frac{\mathfrak{a}_1(\mu_i | \{\lambda\}) - 1}{\mathfrak{a}_1'(\lambda_i | \{\lambda\})} \det_N \left[ \bar{\rho}(\lambda_j, \mu_k) \right] (1 + O(L^{-\infty})), \quad (65)$$

$$\frac{\det_N \hat{H}_2(\boldsymbol{\mu}, \boldsymbol{\lambda})}{\det_N \mathcal{M}_2(\boldsymbol{\mu}, \boldsymbol{\mu})} = \prod_{i=1}^{N} \frac{\mathfrak{a}_2(\lambda_i | \{\mu\}) - 1}{\mathfrak{a}_2'(\mu_i | \{\mu\})} \det_N \left[ \bar{\rho}(\mu_j, \lambda_k) \right] (1 + O(L^{-\infty})). \quad (66)$$

### 3.3 Product formula for the overlap

The determinant in (62) (or equivalently in (65)-(66)) is a generalised Cauchy determinant. It can easily be computed by means of usual arguments for elliptic functions:

$$\begin{aligned}
\det_N \left[ \bar{\rho}(v_j, \omega_k) \right] &= \left( \frac{\vartheta_1'(0)}{\vartheta_2(0)} \right)^N \det_N \left[ \frac{\vartheta_2(v_j - \omega_k)}{\vartheta_1(v_j - \omega_k)} + \frac{\vartheta_2(v_j + \omega_k)}{\vartheta_1(v_j + \omega_k)} \right] \\
&= \left( \frac{\vartheta_1'(0)}{\vartheta_2(0)} \right)^N \prod_{i=1}^{N} \left[ \vartheta_1(2v_i, q^2) \vartheta_4(2\omega_i, q^2) \right] \\
&\quad \times \frac{\displaystyle\prod_{k<j}^{N} \vartheta_1(v_j + v_k) \vartheta_1(v_j - v_k) \vartheta_1(\omega_k + \omega_j) \vartheta_1(\omega_k - \omega_j)}{\displaystyle\prod_{k=1}^{N} \prod_{j=1}^{N} \vartheta_1(v_j + \omega_k) \vartheta_1(v_j - \omega_k)}. \quad (67)
\end{aligned}$$

This enables us to express the overlap (44) as a simple product. Introducing

$$\chi(u) = \frac{\tau_2(u|\{\mu\})}{\tau_1(u|\{\lambda\})}$$
$$= \frac{[\mathfrak{a}_2(u|\{\mu\}) - 1]}{[\mathfrak{a}_1(u|\{\lambda\}) - 1]} \frac{\sin(u - i\xi_2^- - i\zeta/2)}{\sin(u - i\xi_1^- - i\zeta/2)} \prod_{l=1}^{N} \frac{\mathfrak{s}(u - i\zeta, \mu_l)}{\mathfrak{s}(u - i\zeta, \lambda_l)} \frac{\mathfrak{s}(u, \lambda_l)}{\mathfrak{s}(u, \mu_l)}, \quad (68)$$

to be the ratio of the two transfer matrix eigenvalues, and defining the following regularised theta function

$$\varphi(\lambda) = \frac{\vartheta_1(\lambda)}{\sin \lambda} = 2q^{1/4} \prod_{n=1}^{\infty} (1 - q^{2n} e^{2i\lambda})(1 - q^{2n} e^{-2i\lambda})(1 - q^{2n}), \quad (69)$$

we finally obtain

$$S(\{\lambda\}, \{\mu\}) = \prod_{i=1}^{N} \frac{\chi(\lambda_i)}{\chi(\mu_i)} \prod_{i=1}^{N} \prod_{j=1}^{N} \frac{\varphi(\lambda_i + \lambda_j)\varphi(\lambda_i - \lambda_j)}{\varphi(\lambda_i + \mu_j)\varphi(\lambda_i - \mu_j)} \frac{\varphi(\mu_i + \mu_j)\varphi(\mu_i - \mu_j)}{\varphi(\mu_i + \lambda_j)\varphi(\mu_i - \lambda_j)} + O(L^{-\infty}). \quad (70)$$

This simple product formula is valid for large but finite $L$ and $N$ (we recall that for the ground state $N$ is of order $\frac{L}{2}$), up to exponentially small corrections in $L$. To derive it, we have essentially used here the fact that the Bethe states that we consider are characterised by real Bethe roots, with possibly some boundary root. We have also used the absence of holes in the distribution of real roots.

In the next section, we will compute the thermodynamic limit of this product, taking more specifically into account the properties of the ground states that we consider.

## 4   Taking the thermodynamic limit

In this section, we explain how to take the thermodynamic limit of the product formula (70) obtained for the overlap in the previous section. We have to distinguish several cases according to the configurations of the two sets of ground state Bethe roots $\{\lambda\}$ and $\{\mu\}$, see section 2.3.

### 4.1   The case of real Bethe roots

Let us first present the computation of the overlap (70) in the simplest case in which all Bethe roots are real. According to the description of the ground state recalled in section 2.3, this situation occurs for the following respective configurations of the boundary magnetic fields:

(a) case (A') for both states for $L$ odd: $h^+, h_1^-, h_2^- < h_{cr}^{(1)}$ and $h_1^-, h_2^- < -h_+$;

(b) case (B) for both states for $L$ even, which means that we are in one of the two following situations:

    (b1) $h_1^-, h_2^- < h_{cr}^{(1)} < h^+ < h_{cr}^{(2)}$,

    (b2) $h^+ < h_{cr}^{(1)} < h_1^-, h_2^- < h_{cr}^{(2)}$.

### 4.1.1   General strategy

When considering the thermodynamic limit of (70), we have to deal with factors of the form

$$F_f(\nu|\{\lambda\}, \{\mu\}) = \prod_{j=1}^{N} \frac{f(\nu - \lambda_j) f(\nu + \lambda_j)}{f(\nu - \mu_j) f(\nu + \mu_j)}, \quad (71)$$

in which $f$ is a $\pi$-periodic analytic function which does not vanish on the real axis. The thermodynamic limit of products of the form (71), in this simplest case without complex roots nor holes, can be computed as

$$F_f(v|\{\lambda\}, \{\mu\}) = \exp\left\{-\sum_{j=1}^N \left[\log f(v-\mu_j) + \log f(v+\mu_j) - \log f(v-\lambda_j) - \log f(v+\lambda_j)\right]\right\}$$

$$= \exp\left\{-\frac{L}{\pi}\int_{-\frac{\pi}{2}}^{\frac{\pi}{2}} \log f(v-w)\left[\hat{\xi}'_2(w|\{\lambda\}) - \hat{\xi}'_1(w|\{\mu\})\right] dw + O(L^{-\infty})\right\}, \quad (72)$$

in which we have used the sum-to-integral transform, see Corollary B.1 of [2]. The difference of the two counting functions is itself given as

$$\hat{\xi}_2(w|\{\mu\}) - \hat{\xi}_1(w|\{\lambda\}) = \frac{g_2(w) - g_1(w)}{2L}$$

$$+ \frac{1}{2L}\sum_{k=1}^N \left[\theta(w-\mu_k) + \theta(w+\mu_k) - \theta(w-\lambda_k) - \theta(w+\lambda_k)\right]$$

$$= \frac{1}{L}\hat{\xi}_d(w) + O(L^{-\infty}), \quad (73)$$

in which $\hat{\xi}_d$ satisfies the equation

$$\hat{\xi}_d(\lambda) = \frac{1}{2\pi}\int_{-\frac{\pi}{2}}^{\frac{\pi}{2}} \theta(\lambda-\mu)\hat{\xi}'_d(\mu)\,d\mu + \frac{g_2(\lambda) - g_1(\lambda)}{2}. \quad (74)$$

In the cases we consider here, $\hat{\xi}_1(\frac{\pi}{2}) = -\hat{\xi}_1(-\frac{\pi}{2}) = \hat{\xi}_2(\frac{\pi}{2}) = -\hat{\xi}_2(-\frac{\pi}{2}) = \frac{(N+1)\pi}{L}$, so that $\hat{\xi}_d(\frac{\pi}{2}) = -\hat{\xi}_d(-\frac{\pi}{2}) = 0$. Therefore (74) can alternatively be rewritten as

$$\hat{\xi}_d(\lambda) + \int_{-\frac{\pi}{2}}^{\frac{\pi}{2}} K(\lambda-\mu)\hat{\xi}_d(\mu)\,d\mu = \frac{g_2(\lambda) - g_1(\lambda)}{2}, \quad (75)$$

and (72) as

$$F_f(v|\{\lambda\}, \{\mu\}) = \exp\left[-\frac{1}{\pi}\int_{-\frac{\pi}{2}}^{\frac{\pi}{2}} \frac{f'(\lambda-\mu)}{f(\lambda-\mu)}\hat{\xi}_d(\mu)\,d\mu\right]\left[1 + O(L^{-\infty})\right]. \quad (76)$$

The r.h.s. of (75) is a $2\pi$-periodic odd function:

$$g_2(\lambda) - g_1(\lambda) = g_+(\lambda) - g_+(-\lambda), \quad (77)$$

with

$$g_+(\lambda) = -i\epsilon \log \frac{1 - e^{2i\lambda}(p_2 q)^\epsilon}{1 - e^{2i\lambda}(p_1 q)^\epsilon}, \qquad \text{with} \quad p_j = e^{-2\xi_j^-}, \quad (78)$$

and $\epsilon = 1$ if $|p_j q| < 1$, i.e. if we are in case (a) or (b1), whereas $\epsilon = -1$ if $|p_j q| > 1$, i.e. if we are in case (b2). Hence, the integral equation (75) can be solved in terms of Fourier series, and the solution $\hat{\xi}_d(\lambda)$ of (75) can also be written in the form

$$\hat{\xi}_d(\lambda) = \hat{\xi}_+(\lambda) - \hat{\xi}_+(-\lambda), \quad (79)$$

where $\hat{\xi}_+(\lambda)$ is a power series of $e^{2i\lambda}$ (with only positive powers) which satisfies the integral equation

$$\hat{\xi}_+(\lambda) + \int_{-\frac{\pi}{2}}^{\frac{\pi}{2}} K(\lambda-\mu)\hat{\xi}_+(\mu)\,d\mu = \frac{g_+(\lambda)}{2}. \quad (80)$$

Taking into account the explicit form of the kernel $K(\lambda)$ and its Fourier coefficients, one can rewrite this equation as

$$\hat{\xi}_+(\lambda) + \hat{\xi}_+(\lambda + i\zeta) = \frac{g_+(\lambda)}{2}. \tag{81}$$

Introducing

$$\mathfrak{a}_+(u) = \exp\left[2i\hat{\xi}_+(\lambda)\right], \quad u = e^{2i\lambda}, \tag{82}$$

we get the following functional equation equivalent to the integral equation (80):

$$\mathfrak{a}_+(u)\,\mathfrak{a}_+(uq^2) = \left(\frac{1 - u(p_2 q)^\epsilon}{1 - u(p_1 q)^\epsilon}\right)^\epsilon. \tag{83}$$

The unique solution of this equation can be easily expressed

$$\mathfrak{a}_+(u) = \left(\frac{(u(p_2 q)^\epsilon; q^4)_\infty}{(uq^2(p_2 q)^\epsilon; q^4)_\infty} \frac{(uq^2(p_1 q)^\epsilon; q^4)_\infty}{(u(p_1 q)^\epsilon; q^4)_\infty}\right)^\epsilon, \tag{84}$$

in terms of the the q-Pochhammer symbol

$$(x; \alpha)_\infty = \prod_{n=0}^\infty (1 - x\alpha^n). \tag{85}$$

It remains to report the expression (84) into (76) and to compute the integral, so as to obtain an evaluation of the product (71) in the thermodynamic limit, up to exponentially small corrections in $L$.

In particular, if $f$ is of the form

$$f_{p,\pm}(\lambda) = 1 - p\, e^{\pm 2i\lambda}, \qquad \text{with} \quad |p| < 1, \tag{86}$$

the integral in (76) can easily be computed. Using the representation of $f'_{p,\pm}(\lambda)/f_{p,\pm}(\lambda)$ in Fourier series, we straightforwardly obtain that

$$\exp\left[-\frac{1}{\pi}\int_{-\frac{\pi}{2}}^{\frac{\pi}{2}} \frac{f'_{p,\pm}(\lambda - \mu)}{f_{p,\pm}(\lambda - \mu)} \hat{\xi}_d(\mu)\, d\mu\right] = \mathfrak{a}_+(p\, u^{\pm 1}), \qquad u = e^{2i\lambda}. \tag{87}$$

### 4.1.2 Computation of the overlap

We can now apply the previous strategy so as to compute the overlap (70) in the case where all Bethe roots are real.

In particular, from the infinite product representation of the function $\varphi$ (69) in terms of functions of the form (86), we readily obtain

$$F_\varphi(v|\{\lambda\}, \{\mu\}) = \prod_{j=1}^N \frac{\varphi(v - \lambda_j)\,\varphi(v + \lambda_j)}{\varphi(v - \mu_j)\,\varphi(v + \mu_j)}$$

$$= \left[\prod_{n=1}^\infty \mathfrak{a}_+(q^{2n}u)\,\mathfrak{a}_+(q^{2n}u^{-1})\right]\left[1 + O(L^{-\infty})\right]. \tag{88}$$

Let us now consider the ratio of transfer matrix eigenvalues $\chi(\lambda_j)$ evaluated at a Bethe root $\lambda_j$. Using the Bethe equations $\mathfrak{a}_1(\lambda_i|\{\lambda\}) = 1$ we can express the quantity $\mathfrak{a}_2(\lambda_j|\{\mu\})$ as

$$\mathfrak{a}_2(\lambda_j|\{\mu\}) = \frac{\sin(\lambda_j + i\xi_2^- + i\zeta/2)\,\sin(\lambda_j - i\xi_1^- - i\zeta/2)}{\sin(\lambda_j + i\xi_1^- + i\zeta/2)\,\sin(\lambda_j - i\xi_2^- - i\zeta/2)} \frac{\phi(\lambda_j - i\zeta)}{\phi(\lambda_j + i\zeta)}, \tag{89}$$

in which we have introduced the shortcut notation

$$\phi(v) = \prod_{k=1}^{N} \frac{\sin(v - \lambda_k)\sin(v + \lambda_k)}{\sin(v - \mu_k)\sin(v + \mu_k)}. \tag{90}$$

Therefore (68) takes the following form

$$\chi(\lambda_j) = \frac{\phi'(\lambda_j)}{\mathfrak{a}'_1(\lambda_j|\{\lambda\})} \left[ \frac{\sin(\lambda_j + i\xi_2^- + i\zeta/2)}{\sin(\lambda_j + i\xi_1^- + i\zeta/2)} \frac{1}{\phi(\lambda_j + i\zeta)} - \frac{\sin(\lambda_j - i\xi_2^- - i\zeta/2)}{\sin(\lambda_j - i\xi_1^- - i\zeta/2)} \frac{1}{\phi(\lambda_j - i\zeta)} \right]$$

$$= \frac{\phi'(\lambda_j)}{\mathfrak{a}'_1(\lambda_j|\{\lambda\})} \left[ \frac{(1 - u_j(p_2 q)^{-1})(1 - u_j^{-1}(p_2 q)^{-1})}{(1 - u_j(p_1 q)^{-1})(1 - u_j^{-1}(p_1 q)^{-1})} \right]^{\frac{1-\epsilon}{2}} \left( \frac{p_1}{p_2} \right)^{\frac{\epsilon}{2}}$$

$$\times \left[ \left( \frac{1 - u_j(p_2 q)^\epsilon}{1 - u_j(p_1 q)^\epsilon} \right)^\epsilon \frac{1}{\mathfrak{a}_+(q^2 u_j)} - \left( \frac{1 - u_j^{-1}(p_2 q)^\epsilon}{1 - u_j^{-1}(p_1 q)^\epsilon} \right)^\epsilon \frac{1}{\mathfrak{a}_+(q^2 u_j^{-1})} \right] \left[ 1 + O(L^{-\infty}) \right], \tag{91}$$

in which we have set $u_j = e^{2i\lambda_j}$ and used that

$$\phi(\lambda \pm i\zeta) = F_{f_{q^2,\pm}}(\lambda|\{\lambda\}, \{\mu\}) = \mathfrak{a}_+(q^2 e^{\pm 2i\lambda}) \left[ 1 + O(L^{-\infty}) \right]. \tag{92}$$

This expression can be simplified by means of the functional equation (83), and we obtain

$$\chi(\lambda_j) = \frac{\phi'(\lambda_j)}{2iL\xi'_1(\lambda_j|\{\lambda\})} \left[ \frac{(1 - u_j(p_2 q)^{-1})(1 - u_j^{-1}(p_2 q)^{-1})}{(1 - u_j(p_1 q)^{-1})(1 - u_j^{-1}(p_1 q)^{-1})} \right]^{\frac{1-\epsilon}{2}} \left( \frac{p_1}{p_2} \right)^{\frac{\epsilon}{2}}$$

$$\times \left[ \mathfrak{a}_+(u_j) - \mathfrak{a}_+(u_j^{-1}) \right] \left[ 1 + O(L^{-\infty}) \right]. \tag{93}$$

Let us now consider the function

$$\psi(v) = 2iL \frac{\xi'_1(v|\{\lambda\})}{\phi'(v)}. \tag{94}$$

We can evaluate it by means of similar arguments to those used in [25]. Let us first remark that

$$2\pi \sum_{a=1}^{N} [K(\mu - \lambda_a) - K(\mu + \lambda_a)] \frac{1}{\phi'(\lambda_a)} = i \left[ \frac{1}{\phi(\mu + i\zeta)} - \frac{1}{\phi(\mu - i\zeta)} \right], \tag{95}$$

which can easily be proved by comparing poles and residues of the r.h.s and l.h.s as well as their behaviour at infinity. Replacing as usual the sum over Bethe roots by an integral by means of Proposition B.1 of [2] and using (92), we obtain the following equation for the function $\psi(\lambda)$:

$$\int_{-\frac{\pi}{2}}^{\frac{\pi}{2}} K(\mu - v)\psi(v) \, dv = - \left[ \frac{1}{\mathfrak{a}_+(e^{2i\mu}q^2)} - \frac{1}{\mathfrak{a}_+(e^{-2i\mu}q^2)} \right] \left[ 1 + O(L^{-\infty}) \right]. \tag{96}$$

Using the explicit form of the kernel $K(\lambda)$ and analyticity of the function $1/\mathfrak{a}_+(u)$ inside the unit circle we can easily solve this equation:

$$\psi(\mu) = - \left[ \frac{1}{\mathfrak{a}_+(e^{2i\mu})} - \frac{1}{\mathfrak{a}_+(e^{-2i\mu})} \right] \left[ 1 + O(L^{-\infty}) \right]. \tag{97}$$

Inserting this result into (93) we obtain the following very simple formula for the ratio of transfer matrix eigenvalues evaluated in $\lambda_j$:

$$\chi(\lambda_j) = \left(\frac{p_1}{p_2}\right)^{\frac{\epsilon}{2}} \left[\mathfrak{a}_+(q^{1-\epsilon}u_j)\mathfrak{a}_+(q^{1-\epsilon}u_j^{-1})\right]^\epsilon \left[1+O(L^{-\infty})\right]. \tag{98}$$

Combining this result with (88) we obtain

$$\chi(\lambda_j)F_\varphi(\lambda_j|\{\lambda\},\{\mu\}) = \left(\frac{p_1}{p_2}\right)^{\frac{\epsilon}{2}} \left[\prod_{n=1}^\infty \mathfrak{a}_+(q^{2(n-\epsilon)}u_j)\mathfrak{a}_+(q^{2(n-\epsilon)}u_j^{-1})\right] \left[1+O(L^{-\infty})\right]$$

$$= \left(\frac{p_1}{p_2}\right)^{\frac{\epsilon}{2}} \left[\frac{(u_j q^{2-\epsilon}p_2^\epsilon;q^4)_\infty}{(u_j q^{2-\epsilon}p_1^\epsilon;q^4)_\infty} \frac{(u_j^{-1}q^{2-\epsilon}p_2^\epsilon;q^4)_\infty}{(u_j^{-1}q^{2-\epsilon}p_1^\epsilon;q^4)_\infty}\right]^\epsilon \left[1+O(L^{-\infty})\right], \tag{99}$$

in which we have used (84).

The ratio $\chi(\mu_j)$ of transfer matrix eigenvalues evaluated in $\mu_j$ can be expressed similarly as in (98), by simply replacing $u_j = e^{2i\lambda_j}$ by $v_j = e^{2i\mu_j}$ in the expression. Hence, the overlap is given as a product over $j$ of ratios of the form (99) in terms of $\lambda_j$ and $\mu_j$, that we can again evaluate following what has been done in section 4.1.1, using infinite product representation of the q-Pochhammer symbol in terms of functions of the form (86):

$$S(\{\lambda\},\{\mu\}) = \left[\prod_{n=0}^\infty \frac{\mathfrak{a}_+(p_2^\epsilon q^{2-\epsilon+4n})}{\mathfrak{a}_+(p_1^\epsilon q^{2-\epsilon+4n})}\right]^\epsilon \left[1+O(L^{-\infty})\right]$$

$$= \frac{(p_1^{2\epsilon}q^2;q^4,q^4)_\infty (p_2^{2\epsilon}q^2;q^4,q^4)_\infty ((p_1 p_2)^\epsilon q^4;q^4,q^4)_\infty^2}{(p_1^{2\epsilon}q^4;q^4,q^4)_\infty (p_2^{2\epsilon}q^4;q^4,q^4)_\infty ((p_1 p_2)^\epsilon q^2;q^4,q^4)_\infty^2} + O(L^{-\infty}). \tag{100}$$

We recall that $p_i = e^{-2\xi_i^-}$, $i = 1, 2$, and that $\epsilon = 1$ if we are in case (a) or (b1), whereas $\epsilon = -1$ if we are in case (b2). Here $(x;q_1,q_2)_\infty$ denotes the double q-Pochhammer symbol defined as

$$(x;q_1,q_2)_\infty = \prod_{n_1,n_2=0}^\infty (1-xq_1^{n_1}q_2^{n_2}). \tag{101}$$

## 4.2 Cases with presence of a boundary root

We now explain how the previous analytical computations are modified when one of the two states, or both, are described by a set of Bethe roots with a boundary root of the form (40).

### 4.2.1 Presence of two boundary roots $\lambda_{\mathrm{BR}}^{\sigma_1}$ and $\mu_{\mathrm{BR}}^{\sigma_2}$

When considering a product of the form (71), we have now to take into account the presence of these boundary roots $\lambda_{\mathrm{BR}}^{\sigma_1} = -i(\frac{\zeta}{2}+\xi_1^{\sigma_1}+\epsilon_1^{\sigma_1})$ and $\mu_{\mathrm{BR}}^{\sigma_2} = -i(\frac{\zeta}{2}+\xi_2^{\sigma_2}+\epsilon_2^{\sigma_2})$, with $\sigma_1, \sigma_2 \in \{+,-\}$ and $\epsilon_1^{\sigma_1}, \epsilon_2^{\sigma_2} = O(L^{-\infty})$:

$$F_f(v|\{\lambda\},\{\mu\}) = F_{f,\mathrm{BR}}(v) \exp\left[-\frac{1}{\pi}\int_{-\frac{\pi}{2}}^{\frac{\pi}{2}} \frac{f'(v-\mu)}{f(v-\mu)}\hat{\xi}_d(\mu)\,d\mu\right]\left[1+O(L^{-\infty})\right], \tag{102}$$

where $\hat{\xi}_d$ now satisfies the equation[2]

---

[2]We have here $\hat{\xi}_1(\frac{\pi}{2}) = -\hat{\xi}_1(-\frac{\pi}{2}) = \hat{\xi}_2(\frac{\pi}{2}) = -\hat{\xi}_2(-\frac{\pi}{2}) = \frac{N\pi}{L}$, so that $\hat{\xi}_d(\frac{\pi}{2}) = -\hat{\xi}_d(-\frac{\pi}{2}) = 0$.

$$\hat{\xi}_d(\lambda) + \int_{-\frac{\pi}{2}}^{\frac{\pi}{2}} K(\lambda - \mu)\,\hat{\xi}_d(\mu)\,d\mu = \frac{g_2(\lambda) - g_1(\lambda)}{2}$$

$$+ \frac{1}{2} \sum_{\sigma = \pm} \left[ \theta(\lambda - i\sigma(\tfrac{\zeta}{2} + \xi_2^{\sigma_2})) - \theta(\lambda - i\sigma(\tfrac{\zeta}{2} + \xi_1^{\sigma_1})) \right], \quad (103)$$

and

$$F_{f,\mathrm{BR}}(\nu) = \frac{f(\nu + i\frac{\zeta}{2} + i\xi_1^{\sigma_1}) f(\nu - i\frac{\zeta}{2} - i\xi_1^{\sigma_1})}{f(\nu + i\frac{\zeta}{2} + i\xi_2^{\sigma_2}) f(\nu - i\frac{\zeta}{2} - i\xi_2^{\sigma_2})}. \quad (104)$$

**Case with $\lambda_{\mathrm{BR}}^+$ and $\mu_{\mathrm{BR}}^+$:** Let us first consider the case in which the two boundary roots are $\lambda_{\mathrm{BR}}^+$ and $\mu_{\mathrm{BR}}^+$. This occurs when

$$h_1^-, h_2^- < h^+ \qquad \text{with} \quad |h^+| < h_{\mathrm{cr}}^{(1)} \quad \text{or} \quad h_1^-, h_2^- < h_{\mathrm{cr}}^{(1)} < h_{\mathrm{cr}}^{(2)} < h^+. \quad (105)$$

This case is particularly simple since these two boundary roots coincide up to exponentially small corrections in $L$. In other words, the extra factor $F_{f,\mathrm{BR}}(\nu)$ (104) in (102) is identically equal to 1, whereas the term in the second line of (103) vanishes identically. Hence, the presence of the boundary roots does not induce any changes in the computations with respect to the case with only real roots, and the overlap is given by the expression (100) with $\epsilon = 1$.

**Case with $\lambda_{\mathrm{BR}}^-$ and $\mu_{\mathrm{BR}}^-$:** Let us now consider the case in which the two boundary roots are $\lambda_{\mathrm{BR}}^-$ and $\mu_{\mathrm{BR}}^-$. This occurs when

$$h^+ < h_1^-, h_2^- \qquad \text{with} \quad |h_1^-|, |h_2^-| < h_{\mathrm{cr}}^{(1)} \quad \text{or} \quad h^+ < h_{\mathrm{cr}}^{(1)} < h_{\mathrm{cr}}^{(2)} < h_1^-, h_2^-. \quad (106)$$

We can apply in this case similar arguments as in section 4.1.1 and decompose $\hat{\xi}_d$ as in (79), which leads to the following integral equation for $\hat{\xi}_+$:

$$\hat{\xi}_+(\lambda) + \int_{-\frac{\pi}{2}}^{\frac{\pi}{2}} K(\lambda - \mu)\,\hat{\xi}_+(\mu)\,d\mu = -\frac{i}{2} \log\left( \frac{1 - e^{2i\lambda} p_2 q}{1 - e^{2i\lambda} p_1 q} \frac{1 - e^{2i\lambda} p_2^{-1} q}{1 - e^{2i\lambda} p_1^{-1} q} \frac{1 - e^{2i\lambda} p_2 q^3}{1 - e^{2i\lambda} p_1 q^3} \right). \quad (107)$$

Note that, in the regimes (106) considered here, we have both $|p_j q| < 1$ and $|p_j^{-1} q| < 1$, $j = 1, 2$. This equation is equivalent to

$$\mathfrak{a}_+(u)\,\mathfrak{a}_+(uq^2) = \frac{1 - u p_2 q}{1 - u p_1 q} \frac{1 - u p_2^{-1} q}{1 - u p_1^{-1} q} \frac{1 - u p_2 q^3}{1 - u p_1 q^3}, \quad (108)$$

in which we have defined $\mathfrak{a}_+$ in terms of $\hat{\xi}_+$ as in (84). The unique solution of this equation is given by

$$\mathfrak{a}_+(u) = \frac{(u p_2^{-1} q; q^4)_\infty}{(u p_1^{-1} q; q^4)_\infty} \frac{(u p_1^{-1} q^3; q^4)_\infty}{(u p_2^{-1} q^3; q^4)_\infty} \frac{1 - u p_2 q}{1 - u p_1 q}. \quad (109)$$

For $f$ of the form $f_{p,\pm}$ (86), the identity (87) still holds in terms of the function $\mathfrak{a}_+$ (109). Combining this expression with the extra factor (104) due to the presence of the boundary roots in (102), which in this case takes the form

$$F_{f_{p,\pm},\mathrm{BR}}(\lambda) = \frac{(1 - p p_1 q u^{\pm 1})(1 - p(p_1 q)^{-1} u^{\pm 1})}{(1 - p p_2 q u^{\pm 1})(1 - p(p_2 q)^{-1} u^{\pm 1})}, \quad (110)$$

we obtain

$$F_{f_{p,\pm},\mathrm{BR}}(\lambda)\exp\left[-\frac{1}{\pi}\int_{-\frac{\pi}{2}}^{\frac{\pi}{2}}\frac{f'_{p,\pm}(\lambda-\mu)}{f_{p,\pm}(\lambda-\mu)}\,\hat{\xi}_d(\mu)\,d\mu\right]=\widetilde{\mathfrak{a}}_+(pu^{\pm 1}),\tag{111}$$

with

$$\widetilde{\mathfrak{a}}_+(u)=\frac{(up_2^{-1}q;q^4)_\infty\,(u(p_1q)^{-1};q^4)_\infty}{(up_1^{-1}q;q^4)_\infty\,(u(p_2q)^{-1};q^4)_\infty}.\tag{112}$$

Note that $\widetilde{\mathfrak{a}}_+(u)$ coincides with the function obtained in (84) for $\epsilon=-1$, which therefore satisfies (83) with $\epsilon=-1$.

The rest of the computation is therefore nearly identical to what has been done in section 4.1.2 in the case $\epsilon=-1$. The functions $F_\varphi(\nu|\{\lambda\},\{\mu\})$, $\phi(\lambda\pm i\zeta)$ and the ratio $\chi(\lambda_j)$ are then expressed respectively as in (88), (92) and (93) (for $\epsilon=-1$), in terms of the function $\widetilde{\mathfrak{a}}_+$ (112) (instead of in terms of $\mathfrak{a}_+$ (84) for $\epsilon=-1$)[3].

Finally, the overlap is given by the expression (100) with $\epsilon=-1$.

**Case with $\lambda^+_{\mathbf{BR}}$ and $\mu^-_{\mathbf{BR}}$:**  We finally consider the case in which the set of Bethe roots $\{\lambda\}$ contains the boundary root $\lambda^+_{\mathrm{BR}}$, and the set $\{\mu\}$ the boundary root $\mu^-_{\mathrm{BR}}$. This occurs when

$$h_1^-<h^+<h_2^-,\qquad\text{with}\quad|h^+|,|h_2^-|<h^{(1)}_{\mathrm{cr}}\quad\text{or}\quad|h^+|<h^{(1)}_{\mathrm{cr}}<h^{(2)}_{\mathrm{cr}}<h_1^-.\tag{113}$$

Let us consider in that case the ratio of transfer matrices eigenvalues (68) evaluated at $\lambda^+_{BR}$:

$$\chi(\lambda^+_{BR})=\frac{\mathfrak{a}_2\left(\lambda^+_{BR}|\{\mu\}\right)-1}{\mathfrak{a}'_1\left(\lambda^+_{BR}|\{\lambda\}\right)}\frac{\sin\left(\lambda^+_{BR}-i\xi_2^--i\zeta/2\right)}{\sin\left(\lambda^+_{BR}-i\xi_1^--i\zeta/2\right)}\frac{\phi'(\lambda^+_{BR})}{\phi(\lambda^+_{BR}-i\zeta)}.\tag{114}$$

As mentioned in (57), $\mathfrak{a}_2\left(\lambda|\{\mu\}\right)$ diverges exponentially with the system size if $-\zeta<\Im(\lambda)<0$. However, at $\lambda=\lambda^+_{BR}=-i\xi_+-i\zeta/2$ , this divergence is compensated by the presence of the factor $\sin\left(\lambda+i\xi_++i\zeta/2\right)$, which appears in the Bethe equations for both Hamiltonians $\mathbf{H}_1$ and $\mathbf{H}_2$. It also follows from (64) that $\mathfrak{a}'_1\left(\lambda^+_{BR}|\{\lambda\}\right)=O(L^{+\infty})$. Hence, $\chi(\lambda^+_{BR})$ is of order $O(L^{-\infty})$.

Proceeding with the computations similarly to what has been done in previous sections, we can show that all the other factors in (70) remain finite. Hence, the overlap is of order $O(L^{-\infty})$ in this case.

### 4.2.2  Presence of only one boundary root $\lambda^{\sigma_1}_{\mathbf{BR}}$

We finally consider the case in which only one set of Bethe roots, say $\{\lambda\}$, contains a boundary root $\lambda^{\sigma_1}_{\mathrm{BR}}$, the other set of Bethe roots $\{\mu\}$ involving only real roots. This occurs either when

$$h^+<h_1^-\quad\text{with}\quad h^+<h^{(1)}_{\mathrm{cr}}<h_2^-<h^{(2)}_{\mathrm{cr}}\quad\text{and}\quad h_1^-\in\,]-h^{(1)}_{\mathrm{cr}},h^{(1)}_{\mathrm{cr}}[\,\cup\,]h^{(2)}_{\mathrm{cr}},+\infty[,\tag{115}$$

in which case the boundary root is $\lambda^-_{\mathrm{BR}}$, or when

$$h_1^-<h^+\quad\text{with}\quad|h^+|<h^{(1)}_{\mathrm{cr}}<h_2^-<h^{(2)}_{\mathrm{cr}},\tag{116}$$

in which case the boundary root is $\lambda^+_{\mathrm{BR}}$. Note that in both cases we have

$$|p_1q|<1,\qquad|(p_2q)^{-1}|<1,\qquad\text{and}\quad q^2(p_1^{\sigma_1}q)^{\pm 1}<1,\tag{117}$$

---

[3]To compute the ratio of transfer matrix eigenvalues $\chi(\lambda_j)$, it may be convenient to isolate the contribution of the boundary roots, and to compute separately $\chi(\lambda^-_{\mathrm{BR}})$.

in which we have defined $p_1^{\sigma_1} = e^{-2i\xi_1^{\sigma_1}}$.

When considering a product of the form (71), we can still write

$$F_f(v|\{\lambda\},\{\mu\}) = F_{f,\mathrm{BR}}(v)\exp\left[-\frac{1}{\pi}\int_{-\frac{\pi}{2}}^{\frac{\pi}{2}}\log f(v-w)\hat{\xi}'_d(w)\,dw\right][1+O(L^{-\infty})], \qquad (118)$$

with an extra factor $F_{f,\mathrm{BR}}$ given by

$$F_{f,\mathrm{BR}}(v) = f(v+i\tfrac{\zeta}{2}+i\xi_1^{\sigma_1})f(v-i\tfrac{\zeta}{2}-i\xi_1^{\sigma_1}), \qquad (119)$$

and the difference of the two counting function $\hat{\xi}_d$ satisfies the equation

$$\hat{\xi}_d(\lambda) - \frac{1}{2\pi}\int_{-\frac{\pi}{2}}^{\frac{\pi}{2}}\theta(\lambda-\mu)\hat{\xi}'_d(\mu)\,d\mu = \frac{g_2(\lambda)-g_1(\lambda)}{2} - \frac{1}{2}\sum_{\sigma=\pm}\theta(\lambda-i\sigma(\tfrac{\zeta}{2}+\xi_1^{\sigma_1})). \qquad (120)$$

It is convenient, in that case, to consider the derivative of (120),

$$\hat{\xi}'_d(\lambda) + \int_{-\frac{\pi}{2}}^{\frac{\pi}{2}}K(\lambda-\mu)\hat{\xi}'_d(\mu)\,d\mu = \frac{g'_2(\lambda)-g'_1(\lambda)}{2} - \frac{1}{2}\sum_{\sigma=\pm}\theta'(\lambda-i\sigma(\tfrac{\zeta}{2}+\xi_1^{\sigma_1})), \qquad (121)$$

$\hat{\xi}_d$ being the primitive function of the solution $\xi'_d$ of (121) which vanishes at 0. Note that the right hand side of (121) is a $2\pi$-periodic even function of $\lambda$. Hence, using similar arguments as in section 4.1.1, we can look for the solution of (121) as a Fourier series in the form

$$\hat{\xi}'_d(\lambda) = \hat{\xi}'_0 + \hat{\xi}'_+(\lambda) + \hat{\xi}'_+(-\lambda), \qquad (122)$$

in which $\hat{\xi}'_0$ is a contant and $\hat{\xi}'_+(\lambda)$ is a power series of $e^{2i\lambda}$ with only positive powers, and (121) can be rewritten as

$$\hat{\xi}'_0 + \hat{\xi}'_+(\lambda) + \hat{\xi}'_+(\lambda+i\zeta) = \frac{1}{2}\left[\frac{1+e^{2i\lambda}p_1q}{1-e^{2i\lambda}p_1q} + \frac{1+e^{2i\lambda}(p_2q)^{-1}}{1-e^{2i\lambda}(p_2q)^{-1}} + \sum_{\sigma=\pm1}\frac{1+e^{2i\lambda}q^2(p_1^{\sigma_1}q)^\sigma}{1-e^{2i\lambda}q^2(p_1^{\sigma_1}q)^\sigma}\right]. \qquad (123)$$

It follows from (123) that $\hat{\xi}'_0 = 2$. Hence

$$\hat{\xi}_d(\lambda) = 2\lambda + \hat{\xi}_+(\lambda) - \hat{\xi}_+(-\lambda), \qquad (124)$$

in which $\hat{\xi}_+$ is any primitive function of $\hat{\xi}'_+$. Up to a constant which can conveniently be chosen to be zero, the equation (123) can be integrated as

$$\hat{\xi}_+(\lambda) + \hat{\xi}_+(\lambda+i\zeta) = -\frac{1}{2i}\log\left[(1-e^{2i\lambda}p_1q)(1-e^{2i\lambda}(p_2q)^{-1})\prod_{\sigma=\pm1}(1-e^{2i\lambda}q^2(p_1^{\sigma_1}q)^\sigma)\right], \qquad (125)$$

or equivalently

$$\mathfrak{a}_+(u)\,\mathfrak{a}_+(uq^2) = \frac{1}{1-up_1q}\frac{1}{1-u(p_2q)^{-1}}\frac{1}{1-u(p_1^{\sigma_1})^{-1}q}\frac{1}{1-up_1^{\sigma_1}q^3}, \qquad (126)$$

in which we have defined $\mathfrak{a}_+(u) = \exp[2i\hat{\xi}_+(\lambda)]$, $u = e^{2i\lambda}$.

In particular, if $f$ is of the form (86), we have

$$\exp\left[-\frac{1}{\pi}\int_{-\frac{\pi}{2}}^{\frac{\pi}{2}}\log f_{p,\pm}(v-w)\hat{\xi}'_d(w)\,dw\right] = \exp\left[-\frac{1}{\pi}\int_{-\frac{\pi}{2}}^{\frac{\pi}{2}}\frac{f'_{p,\pm}(v-\mu)}{f_{p,\pm}(v-\mu)}\left[\hat{\xi}_+(\mu)-\hat{\xi}_+(-\mu)\right]d\mu\right]$$
$$= \mathfrak{a}_+(pu^{\pm1}), \qquad (127)$$

in which we have used the decomposition (124)-(126) of $\hat{\xi}_d$.

**Case with $\lambda_{\mathrm{BR}}^-$:**    Let us consider more particularly the case in which the boundary root is $\lambda_{\mathrm{BR}}^-$, i.e. the case (115). Then $p_1^{\sigma_1} = p_1$, and the unique solution of the functional equation (126) is

$$\mathfrak{a}_+(u) = \frac{(up_2^{-1}q; q^4)_\infty}{(up_2^{-1}q^{-1}; q^4)_\infty} \frac{(up_1^{-1}q^3; q^4)_\infty}{(up_1^{-1}q; q^4)_\infty} \frac{1}{1-up_1q}. \tag{128}$$

It follows that

$$\widetilde{F}_{f_{p,\pm},\mathrm{BR}}(\lambda) \exp\left[ -\frac{1}{\pi} \int_{-\frac{\pi}{2}}^{\frac{\pi}{2}} \frac{f'_{p,\pm}(\lambda-\mu)}{f_{p,\pm}(\lambda-\mu)} \hat{\xi}_d(\mu)\, d\mu \right] = \widetilde{\mathfrak{a}}_+(pu^{\pm 1}), \tag{129}$$

with $\widetilde{\mathfrak{a}}_+(u)$ given by (112) and coinciding with the function obtained in (84) for $\epsilon = -1$.

The remaining part of the computation is then similar to what has been done in the other cases, and the overlap can be expressed as (100) for $\epsilon = -1$.

**Case with $\lambda_{\mathrm{BR}}^+$:**    In the case in which the boundary root is $\lambda_{\mathrm{BR}}^+$, i.e. the case (116), one can apply similar arguments as for (113), and one finds that the overlapp vanishes up to exponentially small corrections in $L$.

# 5   The overlap in the thermodynamic limit: summary of the results

We now summarize our final results concerning the value of the overlap in the thermodynamic limit, according to the different configurations of the boundary magnetic fields.

## 5.1   The overlap for a chain with an odd number of sites

The study of the thermodynamic limit of the overlap for $L$ odd is quite simple, since we have only very few cases to distinguish, see section 2.3:

1. If $h^+, h_1^-, h_2^- < h_{\mathrm{cr}}^{(1)}$ and $h_1^-, h_2^- < -h_+$, all the Bethe roots are real and the overlap is given by the expression (100) with $\epsilon = 1$:

$$S(\{\lambda\}, \{\mu\}) = \frac{(p_1^2 q^2; q^4, q^4)_\infty\, (p_2^2 q^2; q^4, q^4)_\infty\, (p_1 p_2 q^4; q^4, q^4)_\infty^2}{(p_1^2 q^4; q^4, q^4)_\infty\, (p_2^2 q^4; q^4, q^4)_\infty\, (p_1 p_2 q^2; q^4, q^4)_\infty^2} + O(L^{-\infty}). \tag{130}$$

2. If $|h^+|, h_1^-, -h_2^- < h_{\mathrm{cr}}^{(1)}$ and $h_1^- < -h_+ < h_2^-$, the two ground states are in sectors of different magnetisation, so that the overlap vanishes identically:

$$S(\{\lambda\}, \{\mu\}) = 0. \tag{131}$$

3. If $h^+, h_1^-, h_2^- > -h_{\mathrm{cr}}^{(1)}$ and $h_1^-, h_2^- > -h_+$, the overlap can be obtained by spin-reversal symmetry from the first case, using that

$$(\sigma^x)^{\otimes N} \mathbf{H}_{h^-, h^+} (\sigma^x)^{\otimes N} = \mathbf{H}_{-h^-, -h^+}. \tag{132}$$

Hence the overlap in this case is simply given by the expression (130) in which we have replaced $p_i$ by $p_i^{-1}$:

$$S(\{\lambda\}, \{\mu\}) = \frac{(p_1^{-2} q^2; q^4, q^4)_\infty\, (p_2^{-2} q^2; q^4, q^4)_\infty\, ((p_1 p_2)^{-1} q^4; q^4, q^4)_\infty^2}{(p_1^{-2} q^4; q^4, q^4)_\infty\, (p_2^{-2} q^4; q^4, q^4)_\infty\, ((p_1 p_2)^{-1} q^2; q^4, q^4)_\infty^2} + O(L^{-\infty}). \tag{133}$$

We recall that in all these expressions we have defined $p_i = e^{-2\xi_i^-}$, $i = 1, 2$.

In Fig. 2, we have plotted our analytical result and compared it with numerical results obtained by exact diagonalisation of the Hamiltonian using the Quspin package [26].

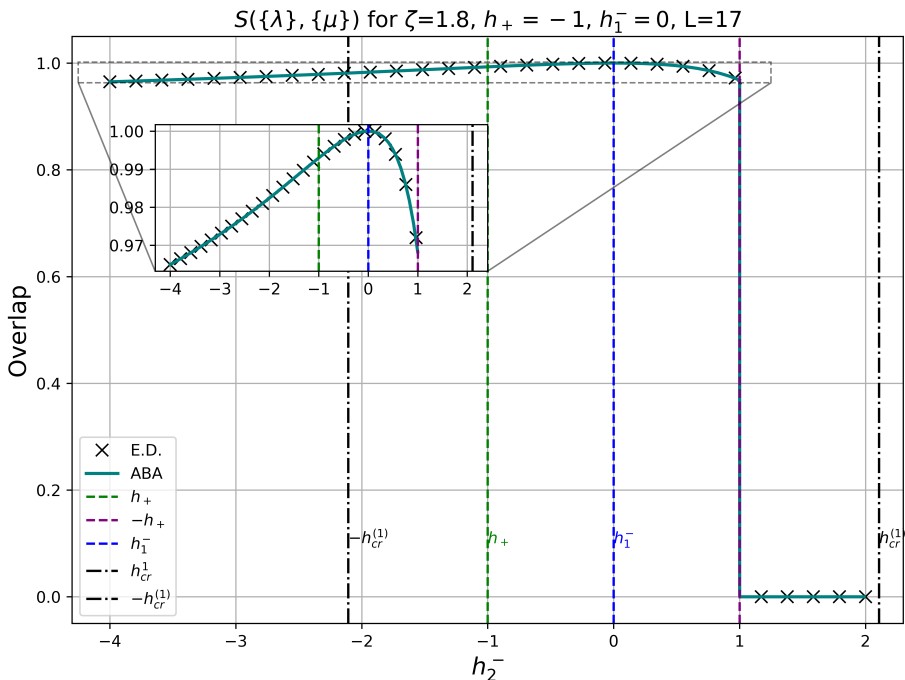

Figure 2: The overlap via exact diagonalisation for a chain of length $L = 17$ compared to the ABA exact result at the thermodynamic limit obtained in section 5.1. Here $\zeta = 1.8$, $h_+ = -1$, $h_1^- = 0$, and the value of the overlap $S(\{\lambda\}, \{\mu\})$ is plotted for different values of $h_2^-$: when $h_2^- < -h_+ = 1$, we are in the case 1 of section 5.1, and the overlap is given by (130); when $h_2^- > -h_+ = 1$, we are in the case 2 of section 5.1 and the overlap vanishes, see (131).

## 5.2 The overlap for a chain with an even number of sites

The analytical study in this case is slightly more complicated than for $L$ odd. Indeed, we have a priori to consider many different configurations for the sets of Bethe roots describing the ground states, according to whether they contain or not a boundary root, see section 2.3. We can nevertheless distinguish the following three main different cases:

1. Case $h_1^-, h_2^- < h^+$. Under the hypothesis that the two ground states are in a configuration (A), (B) or (C), see section 2.3, this may occur in the following situations:

    (i) $h_1^-, h_2^- < h^+$ with $|h^+| < h_{cr}^{(1)}$ or $h_1^-, h_2^- < h_{cr}^{(1)} < h_{cr}^{(2)} < h^+$, i.e. we are in the situation considered in (105).

    (ii) $h_1^-, h_2^- < h_{cr}^{(1)} < h^+ < h_{cr}^{(2)}$, i.e. we are in the situation (b1) considered in section 4.1.

    For all these situations, the overlap is given by the expression (100) with $\epsilon = 1$, i.e.

    $$S(\{\lambda\}, \{\mu\}) = \frac{(p_1^2 q^2; q^4, q^4)_\infty \, (p_2^2 q^2; q^4, q^4)_\infty \, (p_1 p_2 q^4; q^4, q^4)_\infty^2}{(p_1^2 q^4; q^4, q^4)_\infty \, (p_2^2 q^4; q^4, q^4)_\infty \, (p_1 p_2 q^2; q^4, q^4)_\infty^2} + O(L^{-\infty}). \quad (134)$$

2. Case $h_1^- < h^+ < h_2^-$. Under the hypothesis that the two ground states are in a configuration (A), (B) or (C), we may be in one of the two following situations:

(i) $|h^+|, |h_2^-| < h_{\rm cr}^{(1)}$, or $|h^+| < h_{\rm cr}^{(1)} < h_{\rm cr}^{(2)} < h_2^-$, i.e. we are in the situation considered in (113).

(ii) $|h^+| < h_{\rm cr}^{(1)} < h_2^- < h_{\rm cr}^{(2)}$, i.e. we are in the situation considered in (116).

Then the overlap vanishes up to exponentially small corrections in $L$:

$$S(\{\lambda\}, \{\mu\}) = O(L^{-\infty}). \tag{135}$$

3. Case $h^+ < h_1^-, h_2^-$. The overlap can be obtained by spin-reversal symmetry from the first case, i.e.

$$S(\{\lambda\}, \{\mu\}) = \frac{(p_1^{-2}q^2; q^4, q^4)_\infty \, (p_2^{-2}q^2; q^4, q^4)_\infty \, ((p_1 p_2)^{-1}q^4; q^4, q^4)_\infty^2}{(p_1^{-2}q^4; q^4, q^4)_\infty \, (p_2^{-2}q^4; q^4, q^4)_\infty \, ((p_1 p_2)^{-1}q^2; q^4, q^4)_\infty^2} + O(L^{-\infty}). \tag{136}$$

Note that this case corresponds also to several situations in which the overlap has been explicitly computed, namely the situation considered in (106), or in (115), or in case (b2) of section 4.1. For all these situations, the overlap is given by the expression (100) with $\epsilon = -1$, which indeed coincides with (136).

In figures 3 and 4, we compare the analytic expression of the overlap we have obtained in several of these different cases with numerical results obtained by exact diagonalisation using the Quspin package [26].

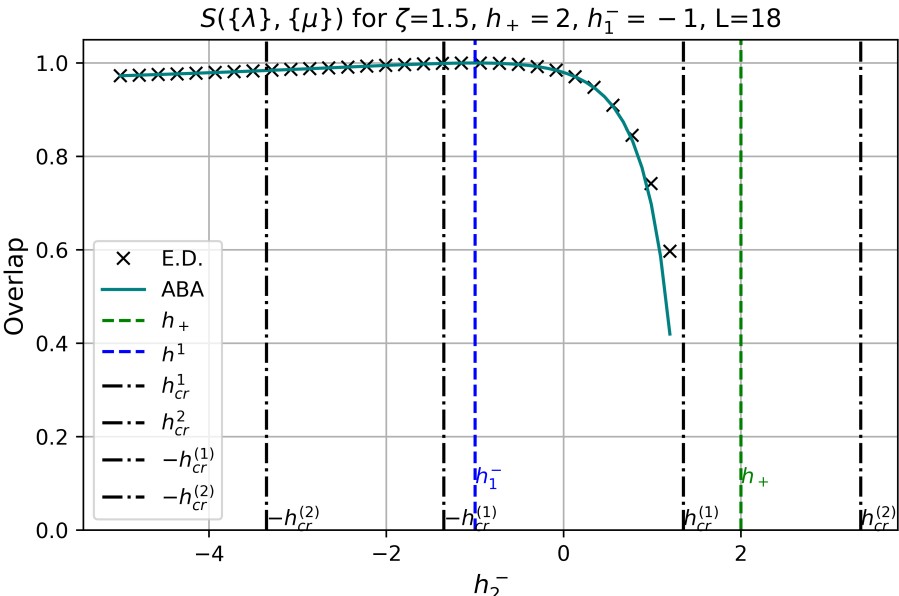

Figure 3: The overlap via exact diagonalisation for a chain of length $L = 18$ compared to the ABA exact result at the thermodynamic limit obtained in section 5.2. Here $\zeta = 1.5$, $h_+ = 2$, $h_1^- = -1$, and the value of the overlap $S(\{\lambda\}, \{\mu\})$ is plotted for different values of $h_2^-$ for $h_2^- < h_{\mathrm{cr}}^{(1)}$: we are here in the configuration (ii) of the case 1 from section 5.2, and the overlap is given by (134).

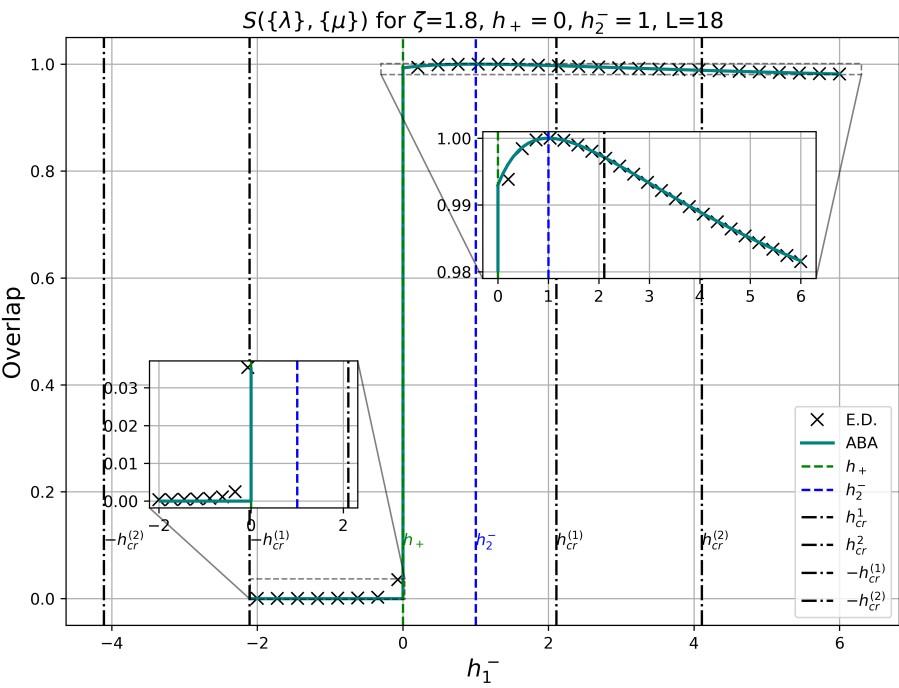

Figure 4: The overlap via exact diagonalisation for a chain of length $L = 18$ compared to the exact result at the thermodynamic limit obtained in section 5.2. Here $\zeta = 1.8$, $h_+ = 0$, $h_2^- = 1$, and the value of the overlap $S(\{\lambda\}, \{\mu\})$ is plotted for different values of $h_1^-$: when $h_1^- < h_+$, we are in configuration (i) of case 2 of the section 5.2 (up to an exchange of $h_1^-$ and $h_2^-$), and the overlap vanishes up to exponentially small corrections in $L$; when $h_1^- > h_+$, we are in case 3 of section 5.2, and the overlap is given by (136).

Note that, as expected, the thermodynamic limit of the overlap does not depend on the parity of the length of the chain, up to a change of $h^+$ into $-h^+$. We would also like to mention that our final expression for the overlap in the thermodynamic limit coincides with a similar result obtained by R. Weston within the framework of the $q$-operator approach [17].

## 6   Conclusion

We have considered a boundary quench in the open XXZ spin chain with boundary magnetic fields parallel to the anisotropy axis, i.e. a change of the value of one of the boundary magnetic fields, and computed the overlaps between the ground states before and after the quench. Our approach is based on the Slavnov determinant representation of the overlaps, and on the Gaudin extraction technique proposed in [14]. In the massive antiferromagnetic regime of the chain $\Delta > 1$, and for all configurations of the magnetic fields for which the spectrum remains gapped (i.e. for which the ground state solutions of the Bethe equations do not include real holes), we have computed the thermodynamic limit of these overlaps up to exponentially small corrections in the length $L$ of the chain.

The fact that we limited our consideration to the massive antiferromagnetic regime enabled us to avoid issues with the convergence of infinite products. It seems however to be possible to apply similar technics in the massless case. In particular, the XXX case can directly be obtained from the massive one as a limit.

In our next publication, we intend to show how to generalise this computation to excited states, and namely how to deal with the presence of real holes. This should effectively open the way to a study of the quench dynamics, or of the behaviour of boundary driven spin chains. Further development of this approach should include all the excited states close to the ground state which implies treatment of the complex roots of the Bethe equations as it was done for the periodic XXX chain [27].

## Acknowledgements

We would like to thank R. Weston for interesting discussions, and in particular for informing us that these kind of results can also be obtained in the framework of the $q$-vertex operator approach [17].

N.K. would like to thank the LPTMS laboratory (Université Paris-Saclay) for hospitality which made this collaboration possible. N.K. is also grateful to the LPTHE laboratory for hospitality.

**Funding information**   The institution of N.K. (IMB) receives support from the EIPHI Graduate School (contract ANR-17-EURE-0002). V.T. is supported by CNRS.

## A   Convergence of numerical results

In this Appendix we illustrate by two plots and corresponding value tables the rapid convergence of numerical results for the overlap toward our analytic formulas (with and without boundary roots).

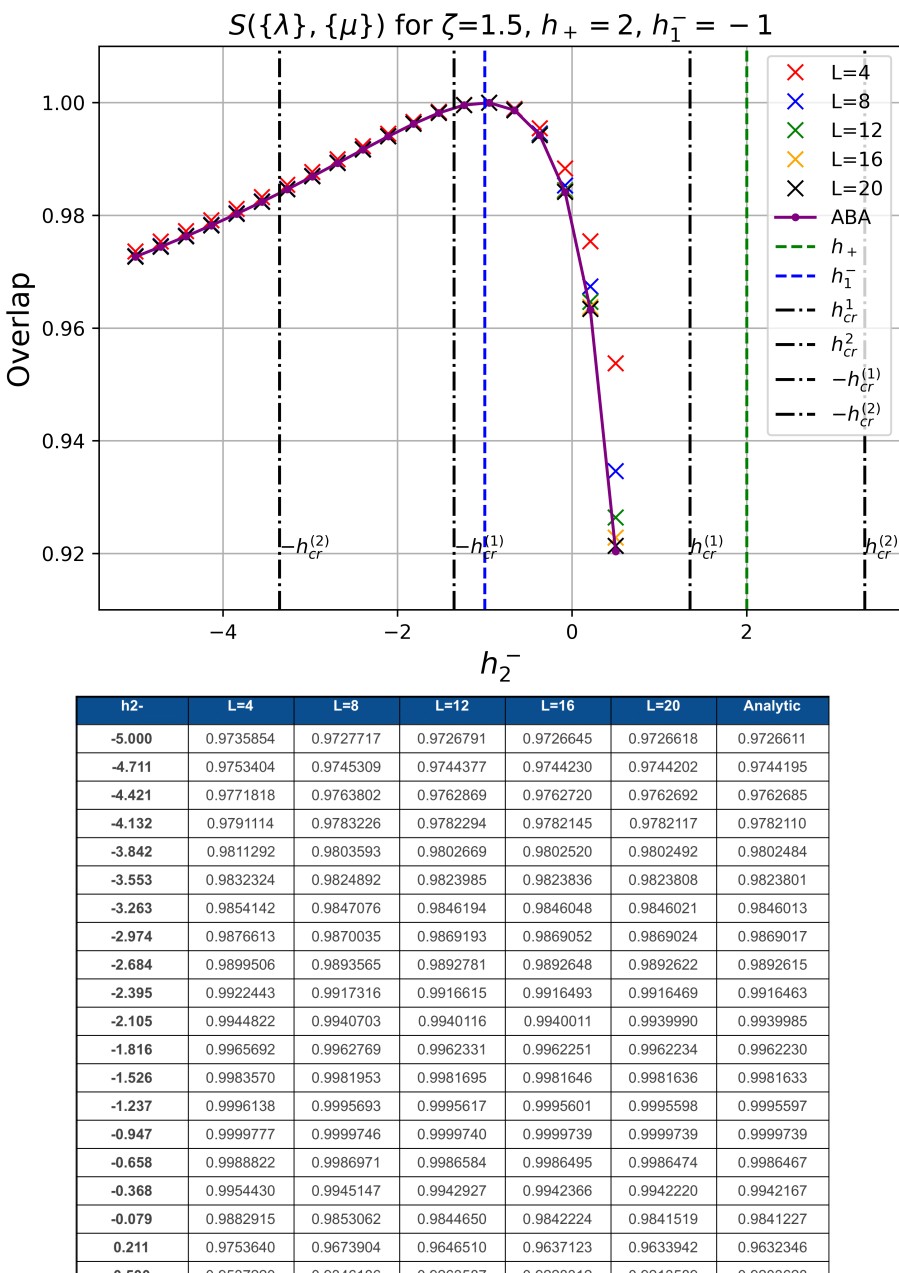

| h2- | L=4 | L=8 | L=12 | L=16 | L=20 | Analytic |
|------|------|------|------|------|------|----------|
| **-5.000** | 0.9735854 | 0.9727717 | 0.9726791 | 0.9726645 | 0.9726618 | 0.9726611 |
| **-4.711** | 0.9753404 | 0.9745309 | 0.9744377 | 0.9744230 | 0.9744202 | 0.9744195 |
| **-4.421** | 0.9771818 | 0.9763802 | 0.9762869 | 0.9762720 | 0.9762692 | 0.9762685 |
| **-4.132** | 0.9791114 | 0.9783226 | 0.9782294 | 0.9782145 | 0.9782117 | 0.9782110 |
| **-3.842** | 0.9811292 | 0.9803593 | 0.9802669 | 0.9802520 | 0.9802492 | 0.9802484 |
| **-3.553** | 0.9832324 | 0.9824892 | 0.9823985 | 0.9823836 | 0.9823808 | 0.9823801 |
| **-3.263** | 0.9854142 | 0.9847076 | 0.9846194 | 0.9846048 | 0.9846021 | 0.9846013 |
| **-2.974** | 0.9876613 | 0.9870035 | 0.9869193 | 0.9869052 | 0.9869024 | 0.9869017 |
| **-2.684** | 0.9899506 | 0.9893565 | 0.9892781 | 0.9892648 | 0.9892622 | 0.9892615 |
| **-2.395** | 0.9922443 | 0.9917316 | 0.9916615 | 0.9916493 | 0.9916469 | 0.9916463 |
| **-2.105** | 0.9944822 | 0.9940703 | 0.9940116 | 0.9940011 | 0.9939990 | 0.9939985 |
| **-1.816** | 0.9965692 | 0.9962769 | 0.9962331 | 0.9962251 | 0.9962234 | 0.9962230 |
| **-1.526** | 0.9983570 | 0.9981953 | 0.9981695 | 0.9981646 | 0.9981636 | 0.9981633 |
| **-1.237** | 0.9996138 | 0.9995693 | 0.9995617 | 0.9995601 | 0.9995598 | 0.9995597 |
| **-0.947** | 0.9999777 | 0.9999746 | 0.9999740 | 0.9999739 | 0.9999739 | 0.9999739 |
| **-0.658** | 0.9988822 | 0.9986971 | 0.9986584 | 0.9986495 | 0.9986474 | 0.9986467 |
| **-0.368** | 0.9954430 | 0.9945147 | 0.9942927 | 0.9942366 | 0.9942220 | 0.9942167 |
| **-0.079** | 0.9882915 | 0.9853062 | 0.9844650 | 0.9842224 | 0.9841519 | 0.9841227 |
| **0.211** | 0.9753640 | 0.9673904 | 0.9646510 | 0.9637123 | 0.9633942 | 0.9632346 |
| **0.500** | 0.9537220 | 0.9346186 | 0.9263587 | 0.9228312 | 0.9213589 | 0.9203628 |

Figure 5: The overlap via exact diagonalization for different chain sizes compared to the exact result from ABA at the thermodynamic limit. Here, the values of the overlap are plotted for $\zeta = 1.5$, $h_+ = 2$ and $h_1^- = -1$. They are also presented in a table to show the rapid (exponential) convergence of the numerical results to the analytic values. In this plot, we are in the exact same configuration as in figure 3.

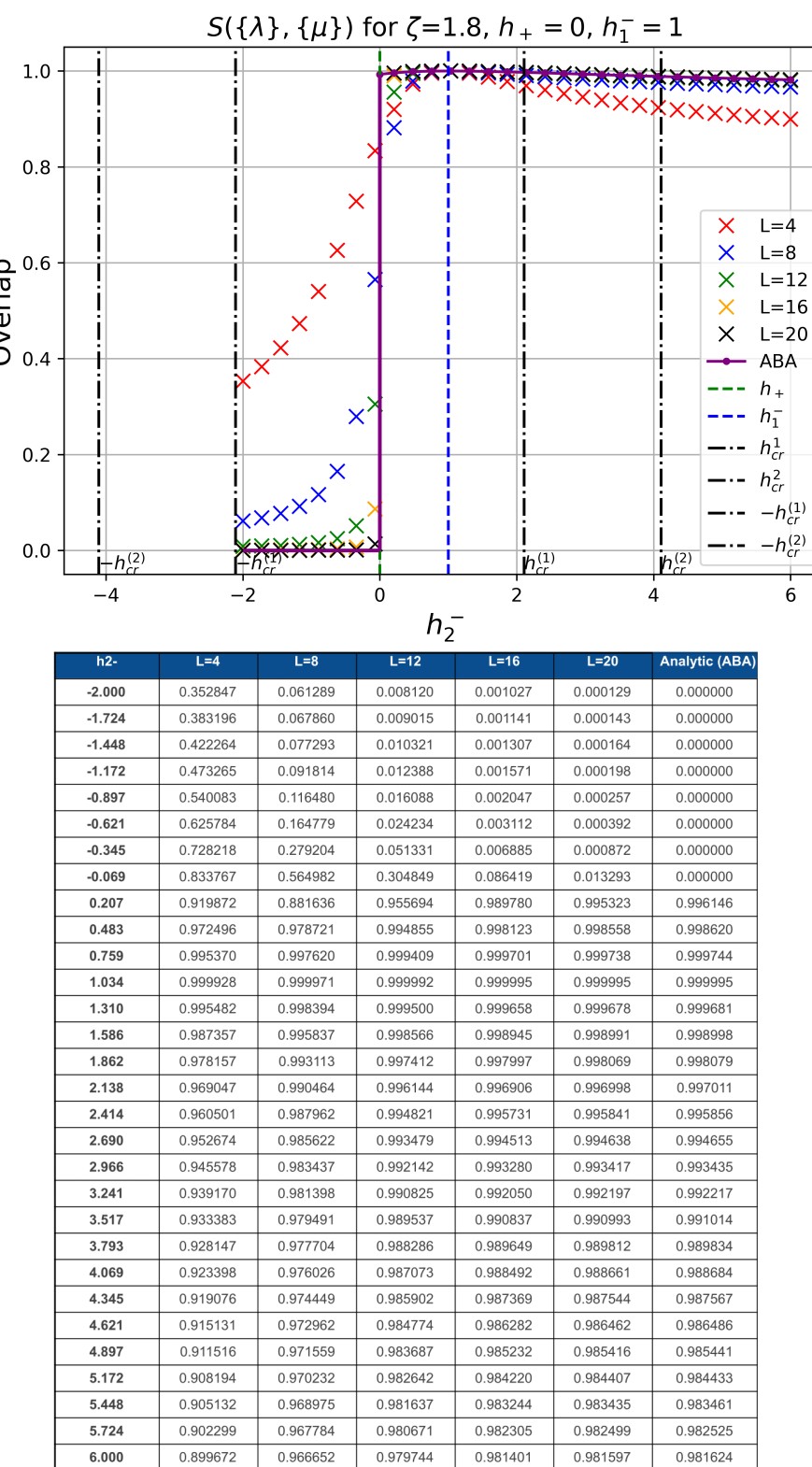

| h2- | L=4 | L=8 | L=12 | L=16 | L=20 | Analytic (ABA) |
|---|---|---|---|---|---|---|
| -2.000 | 0.352847 | 0.061289 | 0.008120 | 0.001027 | 0.000129 | 0.000000 |
| -1.724 | 0.383196 | 0.067860 | 0.009015 | 0.001141 | 0.000143 | 0.000000 |
| -1.448 | 0.422264 | 0.077293 | 0.010321 | 0.001307 | 0.000164 | 0.000000 |
| -1.172 | 0.473265 | 0.091814 | 0.012388 | 0.001571 | 0.000198 | 0.000000 |
| -0.897 | 0.540083 | 0.116480 | 0.016088 | 0.002047 | 0.000257 | 0.000000 |
| -0.621 | 0.625784 | 0.164779 | 0.024234 | 0.003112 | 0.000392 | 0.000000 |
| -0.345 | 0.728218 | 0.279204 | 0.051331 | 0.006885 | 0.000872 | 0.000000 |
| -0.069 | 0.833767 | 0.564982 | 0.304849 | 0.086419 | 0.013293 | 0.000000 |
| 0.207 | 0.919872 | 0.881636 | 0.955694 | 0.989780 | 0.995323 | 0.996146 |
| 0.483 | 0.972496 | 0.978721 | 0.994855 | 0.998123 | 0.998558 | 0.998620 |
| 0.759 | 0.995370 | 0.997620 | 0.999409 | 0.999701 | 0.999738 | 0.999744 |
| 1.034 | 0.999928 | 0.999971 | 0.999992 | 0.999995 | 0.999995 | 0.999995 |
| 1.310 | 0.995482 | 0.998394 | 0.999500 | 0.999658 | 0.999678 | 0.999681 |
| 1.586 | 0.987357 | 0.995837 | 0.998566 | 0.998945 | 0.998991 | 0.998998 |
| 1.862 | 0.978157 | 0.993113 | 0.997412 | 0.997997 | 0.998069 | 0.998079 |
| 2.138 | 0.969047 | 0.990464 | 0.996144 | 0.996906 | 0.996998 | 0.997011 |
| 2.414 | 0.960501 | 0.987962 | 0.994821 | 0.995731 | 0.995841 | 0.995856 |
| 2.690 | 0.952674 | 0.985622 | 0.993479 | 0.994513 | 0.994638 | 0.994655 |
| 2.966 | 0.945578 | 0.983437 | 0.992142 | 0.993280 | 0.993417 | 0.993435 |
| 3.241 | 0.939170 | 0.981398 | 0.990825 | 0.992050 | 0.992197 | 0.992217 |
| 3.517 | 0.933383 | 0.979491 | 0.989537 | 0.990837 | 0.990993 | 0.991014 |
| 3.793 | 0.928147 | 0.977704 | 0.988286 | 0.989649 | 0.989812 | 0.989834 |
| 4.069 | 0.923398 | 0.976026 | 0.987073 | 0.988492 | 0.988661 | 0.988684 |
| 4.345 | 0.919076 | 0.974449 | 0.985902 | 0.987369 | 0.987544 | 0.987567 |
| 4.621 | 0.915131 | 0.972962 | 0.984774 | 0.986282 | 0.986462 | 0.986486 |
| 4.897 | 0.911516 | 0.971559 | 0.983687 | 0.985232 | 0.985416 | 0.985441 |
| 5.172 | 0.908194 | 0.970232 | 0.982642 | 0.984220 | 0.984407 | 0.984433 |
| 5.448 | 0.905132 | 0.968975 | 0.981637 | 0.983244 | 0.983435 | 0.983461 |
| 5.724 | 0.902299 | 0.967784 | 0.980671 | 0.982305 | 0.982499 | 0.982525 |
| 6.000 | 0.899672 | 0.966652 | 0.979744 | 0.981401 | 0.981597 | 0.981624 |

Figure 6: The overlap via exact diagonalization for different chain sizes compared to the exact result from ABA at the thermodynamic limit. Here, the values of the overlap are plotted for $\zeta = 1.8$, $h_+ = 0$ and $h_1^- = 1$. They are also presented in a table to show the rapid (exponential) convergence of the numerical results to the analytic values. In this plot, we are in the same exact configuration as in figure 4.

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
