# Peer review of "Boundary overlap in the open XXZ spin chain"

_SciPost Physics, doi:SciPost Phys. 18, 026 (2025)_

## Round 2 · Referee Report · Sascha Gehrmann (Referee 1) · 2024-11-1

Strengths

1- Opens up a new pathway to study boundary quench dynamics 2-Written in a well-understandable way 3-A detailed case-by-case study for different Bethe roots configurations of the ground state is performed.

Weaknesses

1-Presentation of the accuracy of the numerical data

Report

The paper starts by pointing out that the boundary monodromy matrix used in the algebraic Bethe ansatz framework for an open integrable quantum system only depends on one of the two Skylanin reflection algebras. The authors utilized this fact to study the overlap between Bethe states subject to different boundary conditions (BCs) on one edge of the system. Despite the fact that the different Bethe states belong per se to different integrable models, the invariance of the boundary monodromy under a change of the BCs on one edge allows the employment of known results and recent methods, such as the Slavnov formula or the Gaudin extraction [14], to study their scalar product. The idea is both appealingly simple and interesting from the point of view of later applications, in particular quench dynamics.

The paper focuses on the specific example of the gapped antiferromagnetic XXZ model with two diagonal boundary magnetic fields. In particular, the overlap between the ground states of the system before and after a quench of one edge magnetic field is studied. The study encompasses different parametric regimes of the boundary phase space found in [15,19] and includes the analysis of ground states possessing a boundary root [2]. The main results of the paper are analytical expressions of the ground state overlaps in the thermodynamic limit. The derived results are compared with numerics for small system sizes.

The calculations in the paper seem to be technically correct. The methodology combines traditional approaches and the Gaudin extraction developed in [14]. The paper essentially generalizes the latter to the open case. The documentation of the technical calculations is good and detailed without spoiling the word flow. The main results of the non-vanishing ground state overlaps in the thermodynamic limit are given in terms of (double) $q$-Pochhammer symbols whose numerical evaluation is effective and straightforward. Unfortunately, the excited states are not yet included in the analysis, which would enable a variety of direct applications to boundary quench dynamics. However, the authors claim to report on this later on.

In the end, the analytical results of the overlaps are compared with numerical results from exact diagonalization for small system sizes. The data is represented in Figure 2-4. Despite the notable effort of the authors to present the data adequately by incorporating magnified regions in the graphs, it is not visible from the graphs how good the match is. One cannot directly see the number of the matching digits. A quick numerical reproduction of the data in Fig. 3 (where no magnifier is present) shows that the error is for some parameter domain in the order of $10^{-6}$, way too small for the used axis scale. I would recommend to incorporate extra old-fashioned data tables for all the numerical data demonstrating this accuracy or a link to an online repository where the data is stored.
Further, only the numerical data for $L=18$ is shown but in the introduction, the authors say "We also observe numerically a very rapid convergence towards our analytic result.". Hence, the authors might want to include their numerical validation of the exponential decay of the error in the system size by also showing their data for the system sizes $L=10,12,14,16,(18)$ in one graph for one particular parameter configuration.

All in all, I would like to recommend the paper for publication after some minor changes indicated below have been made, which would improve, in my opinion, the paper even more.

Requested changes

For more details/explanations see the report above.

1- Typo: On page 3, the last paragraph of the introduction "we express the overlap AS a"

2-Too late definition of "primed": in eq (26) the "primed" notation for the derivative is used for $\mathfrak{a}'$, however, this notation for the derivative is just introduced later in the paragraph between eq (35) and eq (36). Just add a small sentence directly after (26).

3-Emphasis: Just before eq (30) I would say COMPLEX CONJUGATED pairs $\lambda_j,\bar{\lambda}_j$ instead of just pairs.

4-Typo: After eq (33) "generaLLy"

5-Typo: In equations (38) and (39) only the critical value $h^{(1)}_{cr}$ is used!

6-Typo: After eq (43) correct the index set to ${\lambda_{\mathbf{1}},...}$

7-Emphasis: Add one sentence after eq (70) to emphasize that the number of roots $N$ must be also large, i.e. of order $\sim L$ to pick the ground state.

8- Too late definition of $u$: In equation (69) $u$ should be replaced by $\lambda$ or the rational notation of the spectral parameter must be here introduced, not in eq (82).

9-Typo: Before the equation (89): we can express THE quantity.

10-Typo: In equation (90) the comma needs to be a dot.

11-Add a table demonstrating the accuracy of the analytic thermodynamic results for small system sizes.

12-Demonstrate numerically the rapid convergence.

Recommendation

Ask for minor revision

---

## Round 2 · Referee Report · Anonymous (Referee 2) · 2024-11-6

Report

The authors compute the overlap of ground states of the open XXZ spin chain in its massive antiferromagnetic regime when one of the boundary fields (in the direction of the anisotropy) is changed.

An important technical aspect in the calculation is that the Bethe states of two models related under such a change are generated by generalized creation and annihilation operators satisfying the same (reflection) algebra. This fact and the fact that the ground state of the model is described by mostly real roots of the corresponding Bethe equations allows to express the overlap for a large but finite system as the product formula eq. (70).

Depending on the value(s) of the boundary fields a local magnetization develops at one of the chain ends. This boundary excitation comes with an imaginary 'boundary root' in addition to the real ones parameterizing the ground state of this model. Taking the thermodynamic limit of eq. (70) the authors consider the different possible cases, i.e. parity of the chain length and no. of boundary roots in the ground states depending on the boundary fields. The results are summarized in Sect. 5 of the paper and compared to numerical data for finite chains.

The authors announce an extension to excited states containing holes in the distribution of real roots in a future publication. As far as I understand another type of excitation would be one where the root configuration for the ground state contains an imaginary boundary root but the excitation not. As an optional addition to the paper it would be interesting if the authors could comment on the possibility of computing the corresponding overlaps within the present framework.

The results of the paper are an important first step towards the study of the response of the spin chain to an abrupt change of a boundary field (quench). I recommend publication in SciPost Physics.

Recommendation

Publish (meets expectations and criteria for this Journal)

---

## Round 2 · Referee Report · Anonymous (Referee 3) · 2024-12-1

Strengths

1- First exact results on overlaps for a quench corresponding to an abrupt change of parameter in an integrable model, opening the way for the analytical study of the boundary quench problem. 2- Very clearly written

Weaknesses

1-Limited to a very specific setting (quench of a parameter at one boundary of the system only)

Report

This works presents a computation of the overlaps between ground states of the open XXZ chain in its gapped regime, for different values of the boundary magnetic field at the left boundary. The calculation combines :

  • the algebraic part, where known expressions for the overlaps between off-shell and an on-shell Bethe states can be directly applied to the present case, treating one of the ground states as on-shell and the other as off-shell. The key point here is that the two states can be constructed from the same boundary Yang-Baxter algebra, which is independent of the value of the quenched boundary parameter.

  • the analytical part (following in part the techniques developed by one of the authors in Ref. [14]), which brings the overlaps to an expression amenable to analytic treatment in the thermodynamic limit.

Specializing to various regimes defined by the values of the pre- and post-quench boundary parameters (and corresponding to different characterizations of the ground state in terms of Bethe roots), the authors present their final result, an exact expression of the overlaps in the thermodynamic limit. Depending upon the regime, those may have a finite value, or become exponentially small in the system size $L$. The results are compared with numerical exact diagonalization, showing compelling agreement.

This is a strong and very clearly written work, leading the way for the analytical study of the boundary quench problem (eventually overlaps with excited states shall be needed, which the authors announce as a work in progress). The only weakness of this approach is that is limited to a very specific protocol, where only one boundary parameter is quenched, as changing parameters at both boundaries (let alone, in the bulk) would spoil the whole algebraic part of the construction. However the present quench problem is significant on its own to deserve a complete study, and I recommend this paper for publication in SciPost.

Requested changes

Besides the typos pointed out by a previous report, I spotted only small grammar mistakes : 1- the authors use repeatedly "reflexion algebra", but I think it should be "reflection" 2- p.9 (2nd paragraph), "since the latter DOES not depend on $h^-$ 3- p.10, beginning of Sec. 3.2: a procedure introduceD in [14] 4- before Sec. 5: one finds that the overlap [not overlapp]

Recommendation

Publish (meets expectations and criteria for this Journal)

---

## Round 3 · Author Response

We thank the referees for their careful reading of the manuscript and their valuable comments.
In our revised version, we have implemented the corrections suggested by the referees. In particular, we have added an appendix with additional numerical plots and data tables to show the rapid convergence towards our analytical result and the accuracy of the analytic thermodynamic results for small system sizes, as requested in 11. and 12 by referee 1; we have also added a little comment and a sentence in the introduction to announce this appendix. We have also implemented the little changes 1. to 10. requested by referee 1, with the exception of 5. which is not a typo : the fact that the spectrum becomes gapless (i.e. the presence of holes) depends only on the critical field $h_cr^{(1)}$, as shown in reference [2] and explained in (38)-(39); however, the detailed configuration of the ground state (presence or not of a boundary complex root) depends also on the critical field $h_cr^{(2)}$, as explained just after.
We have also, as suggested by referee 2, added a comment in the conclusion on taking into account complex roots of Bethe equations (other than the fixed boundary roots).
Finally, we have corrected two minor additional misprints.
In our revised version, we have implemented the corrections suggested by the referees. In particular, we have added an appendix with additional numerical plots and data tables to show the rapid convergence towards our analytical result and the accuracy of the analytic thermodynamic results for small system sizes, as requested in 11. and 12 by referee 1; we have also added a little comment and a sentence in the introduction to announce this appendix. We have also implemented the little changes 1. to 10. requested by referee 1, with the exception of 5. which is not a typo : the fact that the spectrum becomes gapless (i.e. the presence of holes) depends only on the critical field $h_cr^{(1)}$, as shown in reference [2] and explained in (38)-(39); however, the detailed configuration of the ground state (presence or not of a boundary complex root) depends also on the critical field $h_cr^{(2)}$, as explained just after.
We have also, as suggested by referee 2, added a comment in the conclusion on taking into account complex roots of Bethe equations (other than the fixed boundary roots).
Finally, we have corrected two minor additional misprints.

---

## Round 3 · List of Changes

- appendix A added to illustrate convergence of numerical results
- added a remark in the conclusion
- several typos corrected

---

## Editorial Decision

published